# MIRROR: Make Your Object-Level Multi-View Generation More Consistent with Training-Free Rectification

**Tianchi Xing** [* 1] **Bonan Li** [* 1] **Congying Han** [1] **Xinmin Qiu** [1] **Zicheng Zhang** [1] **Tiande Guo** [1]

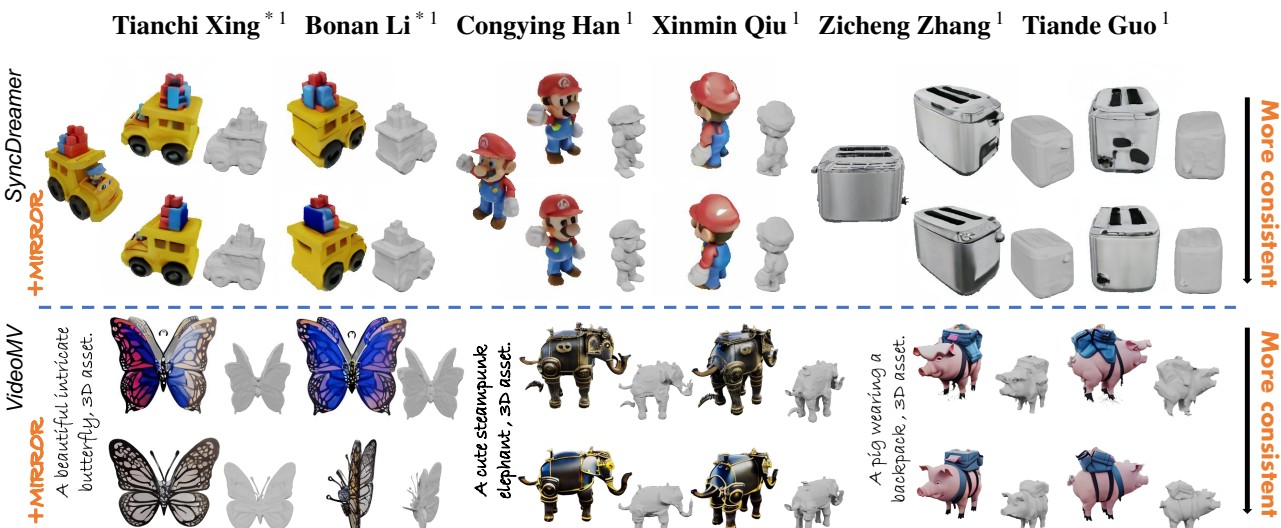

Figure 1: Generated multi-view images by applying MIRROR on SyncDreamer (Liu et al., 2023b) and VideoMV (Zuo et al., 2024) for rectification. MIRROR effectively improves the consistency of generated results in both image-based and text-based multi-view generation models.

## Abstract

Multi-view Diffusion has greatly advanced the development of 3D content creation by generating multiple images from distinct views, achieving remarkable photorealistic results. However, existing works are still vulnerable to inconsistent 3D geometric structures (commonly known as Janus Problem) and severe artifacts. In this paper, we introduce *MIRROR*, a versatile plug-and-play method that rectifies such inconsistencies in a training-free manner, enabling the acquisition of high-fidelity, realistic structures without compromising diversity. Our key idea focuses on tracing the motion trajectory of physical points across adjacent viewpoints, enabling rectifications based on neighboring observations of the same region.

*Equal contribution [1]Department of Mathematical Sciences, University of Chinese Academy of Sciences, No. 19A Yuquan Road, Shijingshan District, Beijing, China. Correspondence to: Congying Han <hancy@ucas.ac.cn>.

*Proceedings of the 42nd International Conference on Machine Learning*, Vancouver, Canada, PMLR 267, 2025. Copyright 2025 by the author(s).

Technically, MIRROR comprises two core modules: **Trajectory Tracking Module** (TTM) for pixel-wise trajectory tracking that labels identical points across views, and **Feature Rectification Module** (FRM) for explicitly adjustment of each pixel embedding on noisy synthesized images by minimizing the distance to corresponding block features in neighboring views, thereby achieving consistent outputs. Extensive evaluations demonstrate that MIRROR, through seamless integration with a variety of off-the-shelf object-level multi-view diffusion models, achieves efficient improvements in consistency and fidelity, thereby functioning as a practical rectification tool.

## 1. Introduction

Recently, 3D asset generation has become one of the most promising fields in computer vision, and demonstrated extensive application in artworks and creation. Inspired by the success in text-to-image generation (Rombach et al., 2022), text-to-3D generation models have increasingly attracted significant attention. Regarding the scarcity of 3D data, the

seminal works leverages 2D diffusion models as priors to optimize 3D representations using score distillation (Poole et al., 2022; Liu et al., 2023a; Wang et al., 2023; Lin et al., 2023; Qian et al., 2023; Wang et al., 2024b). Unfortunately, these methods suffer from serious inconsistencies, most notably the Janus Problem and content drifting, due to the lack of 3D prior knowledge. To address these issues, MV-Dream (Shi et al., 2023b) was the first to introduce the multi-view diffusion model, leveraging joint training on 2D and 3D data to simultaneously generate four orthogonal views. Building upon it, multi-view diffusion models such as SyncDreamer (Liu et al., 2023b), MVD-Fusion (Hu et al., 2024) and VideoMV (Zuo et al., 2024) have been proposed to employ strategies for spatial and temporal alignment to enhance multi-view consistency, thereby enabling the simultaneous generation of dense views at the object level.

Despite notable progress in multi-view diffusion models, they continue to struggle with issues of poor quality and inconsistencies across different perspectives as shown in Fig. 1 and Fig. 5. These issues primarily stem from the extensive resources required for training, the incompleteness of training datasets, and inaccurate correspondences between 3D spatial points and their projections across views. Consequently, it is crucial to delve deeper into efficient 3D asset generation method that accurately captures the real correspondences. To this end, we devise an training-free method, namely *MIRROR*, to amend the multiple images produced by multi-view diffusion models during sampling process, thereby providing consistent results to perform high-quality 3D reconstruction. We showcase remarkable rectification results in Fig. 1.

The core of MIRROR lies in modifying the current viewpoint based on the content of adjacent viewpoints, thereby enforcing neighbor-view consistency and naturally deriving whole-view coherence. To achieve effective rectification, we approach this from the following two aspects: (1) *What should we focus on in neighbor views for rectification?* Compared to image-level adjustments, we opt for point-to-point corrections, ensuring that the features of each physical point remain consistent across different viewpoints. The main challenge of our approach lies in effectively tracking the image coordinates of the same physical point across neighbors. To tackle this, Trajectory Tracking Module (TTM) is proposed to successfully track the motion trajectory of physical points as viewpoints change. This strategy enables precise adjustments for each point, selectively targeting relevant features while disregarding irrelevant ones, which is a key advantage of TTM over the epipolar correspondence methods (Ye et al., 2024; Kant et al., 2024; Zhou & Tulsiani, 2023; Li et al., 2024b). (2) *How to perform rectification effectively and efficiently?* We observe that cross-view attentions (Shi et al., 2023b; Li et al., 2024b) inherently struggle to enforce 3D consistency, as they lack

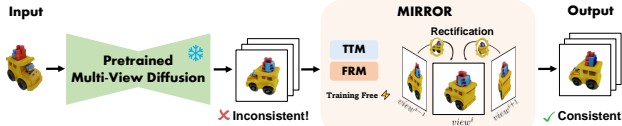

Figure 2: High-level Overview of MIRROR. This is a plug-and-play rectification technique that can be seamlessly incorporated into multi-view diffusion models to enhance consistency and fidelity without any training or fine-tuning.

explicit geometric constraints and rely solely on implicit learning through network weights. Given the limited availability of 3D data, such methods are prone to becoming trapped in local optima and often exhibit poor generalization. Although CoSER (Li et al., 2024a) attempts to address this by injecting local spatial correspondence information from neighboring views into the cross-view attention to promote multi-view consistency, its effectiveness remains limited due to the absence of depth information—which results in inaccurate view correspondences—and the intrinsic challenges in optimizing attention layers as discussed earlier. Therefore, unlike previous approaches that rely on fine-tuning within the attention mechanism, we propose Feature Rectification Module (FRM), which, without requiring any training or fine-tuning, performs feature rectification directly on the predicted images during inference, thereby achieving enhanced performance in a more streamlined and efficient manner. To this end, we compute the similarity between each pixel's feature in the current view and its counterpart in adjacent views during sampling, producing a gradient map that guides denoising. Additionally, to mitigate potential biases and maintain regional continuity, we extend the reference pixel point into a block.

Technically, we propose a two-stage inference strategy. In the first stage, we generate a set of images using the multi-view diffusion model to extract depth information for each view. Based on transformations in camera angles, in TTM, we introduce a *trajectory tracking operator* guided by depth maps to track motion trajectories of 3D physical points from the current view to adjacent views at the fixed elevation. This enables precise localization of the corresponding points in neighboring views. We then utilize DDIM Inversion (Song et al., 2020) to progressively add noise to the generated images, transitioning them to intermediate noisy states for the subsequent denoising process. During each sampling step of the second-stage inference, we apply a corrective gradient map based on pixel-level similarity in FRM to refine the denoising direction of the multi-view diffusion model, resulting in images with improved quality and consistency. To address background redundancy caused by surrounding blocks, we implement a dual-anchor feature fusion strategy, which fuses relevant features from adjacent views. This not only identifies meaningful features but also significantly improves computational efficiency.

Furthermore, gradient-based guidance, as discussed in the prior work (Dhariwal & Nichol, 2021), is known to be computationally intensive due to the gradient computation requirements for large neural networks. To mitigate this, we provide a theoretical analysis demonstrating that omitting the UNet Jacobian term in the diffusion model during gradient-based denoising guidance introduces negligible error. Experimentally, we validate that this optimization achieves a twofold acceleration in the rectification process without compromising the quality of the generated images. An overview of MIRROR is depicted in Fig. 2.

In summary, our contributions are as follows:

- We innovatively introduce MIRROR, an efficient, training-free, plug-and-play method that enhances multi-view consistency in 3D asset generation by directly rectifying latent noisy images across views during the sampling process.

- We present TTM based on depth information to precisely ascertain the corresponding positions of points across distinct views, and FRM to eliminate ambiguity by enforcing consistency in the representation of the same physical point across different viewpoints.

- Extensive experiments demonstrate that MIRROR consistently enhances the performance of various multi-view generator, both quantitatively and qualitatively.

## 2. Related Work

### 2.1. 3D Generation

Due to the scarcity of 3D data, numerous studies have turned to high-quality 2D diffusion models for 3D generation tasks. A fundamental technique in this domain, Score Distillation Sampling (SDS), introduced by DreamFusion (Poole et al., 2022), utilizes pre-trained 2D models as priors to optimize 3D representations, enabling zero-shot text-to-3D generation without 3D data. Inspired by it, several works adopt this pipeline to optimize a neural radiance field (Mildenhall et al., 2021) to generate 3D assets (Lin et al., 2023; Liu et al., 2023a; Qian et al., 2023; Wang et al., 2024b; Zhu et al., 2023; Huang et al., 2023). Furthermore, DreamGaussian (Tang et al., 2023) and GaussianDreamer (Yi et al., 2024) accelerate the optimization process by applying Gaussian Splitting techniques (Kerbl et al., 2023) in Score Distillation Sampling. However, these 2D-lifting methods exhibit critical drawbacks, such as severe multi-view inconsistencies and slow generation speeds, often requiring tens of thousands of iterations to produce a single 3D asset. In response to these challenges, recent research on multi-view diffusion models has emerged as a promising direction in 3D generation, wherein the model is capable of generating multiple views of an object in a single inference process.

### 2.2. Multi-View Diffusion Methods

Recent advancements have extended 2D diffusion models to generate multi-view images for reconstruction. Notable developments include MVDream (Shi et al., 2023b), Viewset Diffusion (Szymanowicz et al., 2023), SyncDreamer (Liu et al., 2023b). However, these methods face challenges like texture ambiguity and high computational costs. To address these, Wonder3D (Long et al., 2024) introduces a cross-domain model for generating normal maps, while Zero123++ (Shi et al., 2023a) and One-2-3-45 (Liu et al., 2024) improve texture quality and accelerate 3D reconstruction. Despite these innovations, quality issues persist due to reliance on limited synthetic data. Video generative models, with their temporal modules ensuring frame consistency, are increasingly favored for multi-view generation over image diffusion models. SV3D (Voleti et al., 2024) and IM-3D (Melas-Kyriazi et al., 2024) enhance 3D generation by leveraging advanced video diffusion models to optimize multi-view synthesis for improved output quality. And VideoMV (Zuo et al., 2024) also develops a consistent multi-view generation model using video generative models, further improving multi-view consistency with a 3D Aware Denoising Sampling technique. While these methods improve consistency, they lack explicit geometric constraints, entirely relying on learned networks. With limited 3D data, they are prone to local optima, and issues like Janus Problem and content drifting remain at inference. Building on these efforts, we propose MIRROR, an efficient, training-free rectification technique specifically designed to enhance 3D consistency in these dense multi-view diffusion models.

## 3. Preliminaries

### 3.1. Diffusion Models

Diffusion Models (DMs) (Ho et al., 2020) learn a target distribution $p_\theta(x_0)$, by progressively denoising a standard Gaussian distribution. This process is mathematically represented as: $p_\theta(x_0) = \int p_\theta(x_{0:T}) \, dx_{1:T}$, where $x_{1:T}$ denotes intermediate noisy samples. In the forward process, DMs iteratively add Gaussian noise $\varepsilon$ to the clean data $x_0$, controlled by a pre-defined variance schedule, formulated as:

$$x_t = \sqrt{\overline{\alpha}_t} x_0 + \sqrt{1 - \overline{\alpha}_t} \varepsilon, \ \varepsilon \sim \mathcal{N}(0, \boldsymbol{I}). \quad (1)$$

Conversely, in the reverse process, DMs employ a denoiser $\varepsilon_\theta$, parameterized as a UNet, to predict the noise added at each denoising time step $t$.

Furthermore, to reduce the computational demands associated with synthesizing high-resolution images, Latent Diffusion Models (LDMs) (Rombach et al., 2022) implement a pre-trained encoder $\mathcal{E}$ of VQGAN to compress image $x_0 \in \mathbb{R}^{H \times W \times C}$ into a low-dimensional latent feature, $z_0 = \mathcal{E}(x_0)$, where $z_0 \in \mathbb{R}^{h \times w \times c}$. In addition, LDMs uti-

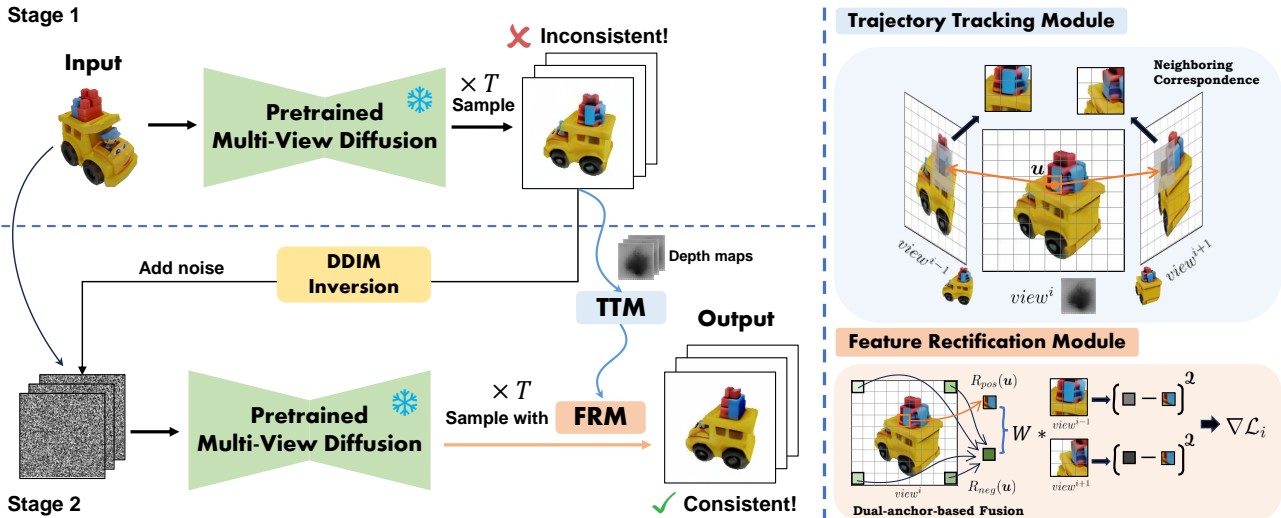

Figure 3: Pipeline of MIRROR. We employ a two-stage sampling strategy. **In stage 1**, given a single image or a text prompt as input, we employ a base multi-view diffusion model to sample a set of images. **In stage 2**, we first apply DDIM Inversion to convert generated images into the noisy state, and take it as the initial noise for the second stage. Then, Trajectory Tracking Module (TTM) is employed to capture the motion trajectory of physical points from the current view to the adjacent views using the depth maps derived from generated images in stage 1. For Feature Rectification Module (FRM), the block features are fused based on dual anchors, $R_{pos}(\boldsymbol{u})$ and $R_{neg}(\boldsymbol{u})$. Then, we explicitly enforce consistency for the same physical point across neighbor views by calculating the distance between the feature of point $\boldsymbol{u}$ and the corresponding block features in adjacent views, and compute the overall gradient map $\nabla\mathcal{L}_t$ to guide the denoising process for rectification.

lize deterministic DDIM sampling (Song et al., 2020) to efficiently transform random noise $z_T$ into clean data $z_0$ through a sequence of discrete time steps from $t = T$ to $t = 1$, which is formulated as:

$$z_{t-1} = \sqrt{\overline{\alpha}_{t-1}} \left( \frac{z_t - \sqrt{1 - \overline{\alpha}_t}\, \varepsilon_\theta(z_t, t)}{\sqrt{\overline{\alpha}_t}} \right) \\ + \sqrt{1 - \overline{\alpha}_{t-1}}\, \varepsilon_\theta(z_t, t). \quad (2)$$

This sequence ultimately results in the recovery of the latent clean state $z_0$, which is then processed by the VQGAN decoder $\mathcal{D}$ to reconstruct a high-fidelity image $x_0 = \mathcal{D}(z_0)$.

Evolved from classical LDMs (Rombach et al., 2022), Multi-View Diffusion Models (MV-DMs) have broadened their focus from single images to handling sequences of multi-view images, essentially treating them like videos. Specifically, consider an image sequence $x_0$ represented in $\mathbb{R}^{F \times H \times W \times 3}$, where $F$ denotes the number of frames. MV-DMs encode this video into a latent space, represented as $z_0 = \mathcal{E}(x_0)$, where $z_0 \in \mathbb{R}^{F \times h \times w \times c}$.

### 3.2. DDIM Inversion

In contrast to DDIM, DDIM inversion acts as a forward process, where clean data $z_0$ progressively transitions into a noisy state $z_T$, effectively reversing the original sampling

sequence Eq. (2) into:

$$z_{t+1} = \sqrt{\overline{\alpha}_{t+1}} \left( \frac{z_t - \sqrt{1 - \overline{\alpha}_t}\, \varepsilon_\theta(z_t, t)}{\sqrt{\overline{\alpha}_t}} \right) \\ + \sqrt{1 - \overline{\alpha}_{t+1}}\, \varepsilon_\theta(z_t, t). \quad (3)$$

## 4. Methods

In this section, we propose **MIRROR**, a training-free modifier specifically designed to enhance consistency across images generated by multi-view diffusion models. In Sec. 4.1, we systematically outline our approach. Then, we introduce Trajectory Tracking Module (TTM) that tracks point transformation correspondences between adjacent views (Sec. 4.2) and proposes a Feature Rectification Module (FRM) that integrates 3D-aware feature information to rectify inconsistencies during the sampling process (Sec. 4.3).

### 4.1. Framework of MIRROR

Given a pre-trained muti-view diffsuion, we adopt a two-stage sampling strategy to rectify generated images. In the first stage, the base multi-view model generates a set of images $\hat{x}_0$, from which the depth map $\boldsymbol{Z}'_{absolute}$ is derived via monocular depth estimation. Subsequently, DDIM Inversion is then applied to convert clean images into noisy state $z_T$, serving as the initial noise for the second stage of the sampling process. At each denoising step $t$, TTM

tracks point correspondences of the predicted latent images, $\hat{z}_0 = \hat{z}_0(z_t; t)$, across adjacent views with $\boldsymbol{Z}'_{absolute}$. Given this, FRM integrates 3D-aware feature information according to the correspondences from TTM, and computes a pixel-level rectification loss gradient $\nabla \mathcal{L}_t$ to refine the denoising direction of the predicted noise $\varepsilon_\theta$. Finally, $z_{t-1}$ is generated through DDIM sampling with the rectified noise $\hat{\varepsilon}_\theta$, enhancing the quality and consistency of the final output. The pipeline of MIRROR is illustrated in Fig. 3, with the corresponding pseudocode provided in Algorithm 1.

## 4.2. Trajectory Tracking Module

***Pixel-Wise Trajectory Tracking Transformation.*** In theory, there exists a pixel-wise correspondence between two images of an object projected from different viewpoints. We observe that large viewpoint gaps under arbitrary camera trajectories tend to reduce cross-view correlation, resulting in increased reconstruction errors and limited improvement. In contrast, fixing the elevation angle strengthens inter-view correspondence and yields better results by providing more related information. Therefore, we adopt a standardized setting with a fixed elevation angle. In this setup, we consider $F$ views captured by a camera rotating horizontally around the object in a standardized manner. The circular camera trajectories ensure stable and uniform coverage of the viewing space, thereby enhancing view correlation and improving the quality of 3D reconstruction. Furthermore, sampling across multiple elevation settings helps recover occluded regions that are invisible from a single elevation. Specifically, this configuration maintains a constant elevation angle for the images while distributing the azimuth angles uniformly across 360°. Thus, let $\alpha$ represent the change in azimuth angle between each pair of adjacent views.

Next, we establish the pixel-wise correspondence on the imaging plane between two differing viewpoints. As the camera angle variation is confined solely to the azimuth direction, this motion can be modeled as the imaging plane rotating around the object along the elevation axis. Consequently, the $y$-coordinate (height) remains constant, while the $x$-coordinate (width) undergoes a rotation transformation related to the $z$-coordinate (depth). Let $V$ be the function space $L^2(\mathbb{R}^2, \mathbb{R})$, the specific transformation formula defined on $V$ is outlined in Definition 4.1.

**Definition 4.1.** The trajectory tracking operator $\mathcal{T}_\alpha : V \to V$, defined by an azimuthal rotation angle $\alpha \in \mathbb{R}$, is formulated as $\mathcal{T}_\alpha(Z(\boldsymbol{u})) = Z_\alpha(\boldsymbol{u}_\alpha)$, where $\boldsymbol{u} = (x, y)$, $\boldsymbol{u}_\alpha = (x_\alpha, y_\alpha)$. The coordinates transformation is given by

$$\begin{cases} x_\alpha = (x - \dfrac{W}{2}) \cos \alpha + \dfrac{W}{2} - z \sin \alpha, \\ y_\alpha = y, \end{cases} \quad (4)$$

where $z$ represents the depth (distance from the camera center to the object), and $W$ is the width of the image.

***Depth Prior Estimation.*** Since images directly output by multi-view dffusion models typically lack depth information, we utilize a monocular depth estimation model $\mathcal{H}$ to estimate this crucial detail $z$ in Eq. (4). However, given the potential for considerable bias in depth estimates derived from noisy images, we feed the clean images $\hat{x}_0$ that have been denoised by the multi-view diffusion model into $\mathcal{H}$, enabling us to accurately assess depth information for each viewpoint, ultimately leading to the creation of a relative normalized depth map $\mathcal{Z}_{relative} := \mathcal{H}(\hat{x}_0) \in \mathbb{R}^{F \times H \times W}$. Although the depth is estimated from a monocular model, the depth scale factor $r$ remains consistent across views, as each baseline multi-view diffusion model fixes the camera distance under circular poses during training. This consistency enables the transformation of relative depth into absolute depth for cross-view alignment, formulated as $\mathcal{Z}_{absolute} = r \cdot \mathcal{Z}_{relative}$.

Furthermore, we establish that as the denoising step $t$ decreases, the trajectory tracking of the predicted images $\hat{x}_0(x_t, t) = \frac{1}{\sqrt{\overline{\alpha}_t}}(x_t - \sqrt{1 - \overline{\alpha}_t}\, \varepsilon_\theta(x_t, t))$ at time step $t$ converges to that of the real images $x_0$, which is detailed in the following proposition.

**Proposition 4.2.** *(Convergence Analysis). The trajectory tracking error of $\mathcal{T}_\alpha$ is upper-bounded by the error induced by the diffusion model, which is given by*

$$\|\mathcal{T}_\alpha(\hat{x}_0(x_t, t)) - \mathcal{T}_\alpha(x_0)\| \le O(\|\hat{x}_0(x_t, t) - x_0\|), \quad (5)$$

*where $\|\cdot\|$ is the $L_2$ norm and $O$ denotes a term of the same infinitesimal order. Moreover, it converges to zero as $t \to 0$.*

The proof is provided in Appendix B.1. Next, to leverage TTM during sampling for guiding the denoising direction via pixel-level correspondence, the depth map must be interpolated to match the resolution of the latent space, resulting in: $\mathcal{Z}'_{absolute} = \mathcal{B}(\mathcal{Z}_{absolute}, h, w)$, where $\mathcal{B}$ represents the bilinear interpolation function. And the visualization results of trajectory tracking based on depth information are provided in Appendix G. Unlike methods (Ye et al., 2024; Kant et al., 2024; Zhou & Tulsiani, 2023; Li et al., 2024b) that integrate epipolar geometry into multi-view attention, MIRROR uniquely employs a more precise and efficient point-to-point correspondence tracking strategy. As shown in Fig. 4, we demonstrate that epipolar correspondence establishes an inaccurate relationship, thus introduces irrelevant background features, leading to overly smooth rectification that lacks detail and the Janus problem. In contrast, MIRROR-rectified images maintain higher fidelity and visual quality.

## 4.3. Feature Rectification Module

***Feature Scrutiny.*** To maintain the continuity of pixel-wise feature information and in consideration of potential biases

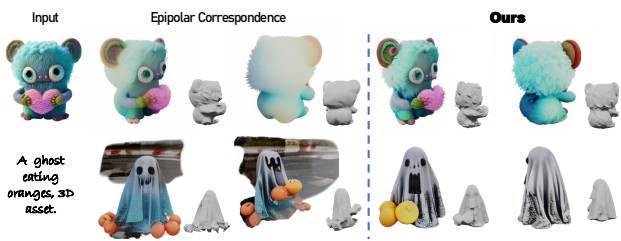

Figure 4: Comparison of the rectification results between the epipolar correspondence method and MIRROR. For better presentation, the background of each image is removed.

in depth information estimation, we adopt a more robust approach, which extracts the most relevant corresponding features from adjacent views, thereby delivering a more precise direction for consistency rectification. Specifically, for each spatial point $\boldsymbol{u}$ in the current view, we select not only its corresponding point $\boldsymbol{u}_\alpha$, but also the surrounding $3 \times 3$ block in adjacent views, making a total of nine points, denoted as $B_{\boldsymbol{u}_\alpha}$, as the feature interaction block for point $\boldsymbol{u}$. This method is figuratively described as the "looking forward and back" feature scrutiny approach, emphasizing a comprehensive examination of the contextual area.

***Feature Information Fusion.*** As mentioned in the introduction, we directly utilize the predicted map $\hat{z}_0 = \hat{z}_0(z_t; t)$:

$$\hat{z}_0 = \frac{z_t - \sqrt{1 - \overline{\alpha}_t}\, \varepsilon_\theta(z_t, t)}{\sqrt{\overline{\alpha}_t}}, \hat{z}_0 \in \mathbb{R}^{F \times h \times w \times c}, \quad (6)$$

from each reverse step $t$ as feature tensors, rather than extracting features from the intermediate layers of the UNet network. Our approach offers a simpler and more effective method of feature extraction, providing superior pixel-level alignment at the same time. Since the interaction block $B_{\boldsymbol{u}_\alpha}$ may contain invalid background information, we employ a feature information fusion method based on both positive and negative anchors. The target correction point $\boldsymbol{u}$ of the current view is selected as the positive anchor $R_{pos}(\boldsymbol{u})$, while the mean of the four corners serves as the negative anchor $R_{neg}(\boldsymbol{u})$. We calculate the cosine similarity between points in the feature interaction block $B_{\boldsymbol{u}_\alpha}$ and dual anchors, $R_{pos}(\boldsymbol{u})$ and $R_{neg}(\boldsymbol{u})$. Subsequently, compute the weighted map $W$, where higher values indicate closer distance to the positive anchor $R_{pos}(\boldsymbol{u})$ and greater distance from the negative anchor $R_{neg}(\boldsymbol{u})$:

$$W = m^+ \mathrm{softmax}\left(\mathcal{C}^+(\boldsymbol{u}, \boldsymbol{v})\right) + m^- \mathrm{softmax}\left(\mathcal{C}^-(\boldsymbol{u}, \boldsymbol{v})\right), \quad (7)$$

for all $\boldsymbol{v} \in B_{\boldsymbol{u}_\alpha}$, where $m^\pm$ denote the combined coefficients satisfying $m^+ + m^- = 1$, and $\mathcal{C}^\pm(\boldsymbol{u}, \boldsymbol{v})$ refer to

$$\mathcal{C}^+(\boldsymbol{u}, \boldsymbol{v}) = \mathrm{Cosine\ Similarity}\left(R_{pos}(\boldsymbol{u}), \boldsymbol{v}\right), \\ \mathcal{C}^-(\boldsymbol{u}, \boldsymbol{v}) = 1 - \mathrm{Cosine\ Similarity}\left(R_{neg}(\boldsymbol{u}), \boldsymbol{v}\right). \quad (8)$$

Given the weighted map $W$, the feature interaction block

can be fused as a mapping $\mathcal{M} : V \to V$, where $\mathcal{M}$ is defined by

$$\mathcal{M}(Z_\alpha(\boldsymbol{u}_\alpha)) \triangleq \sum_{\boldsymbol{v} \in B_{\boldsymbol{u}_\alpha}} W_{\boldsymbol{v}} \times Z_\alpha(\boldsymbol{v}). \quad (9)$$

This approach is strategically designed to distance the fused feature points from irrelevant background and enhance their relevance to the target correction point $\boldsymbol{u}$.

***Rectification Formulation.*** At first, We can define the adjacent operator $\mathcal{F}_\alpha$ and use its $L_2$-norm to measure the difference between two different views as follows.

**Definition 4.3.** The operator $\mathcal{F}_\alpha : V \to V$ is referred to as the adjacent operator, and is defined as

$$\mathcal{F}_\alpha \triangleq \mathcal{I} - \mathcal{M} \circ \mathcal{T}_\alpha,$$

where $\mathcal{I}$ denotes the identity operator on space $V$, and $\mathcal{M}$ and $\mathcal{T}_\alpha$ is defined in Eq. (9) and Definition 4.1, respectively, with $\circ$ indicating the composition of operators. Note that the composition operator $\mathcal{M} \circ \mathcal{T}$ is a group homomorphism (detailed in Appendix B.4). Then, the correction loss function of the spatial point $\boldsymbol{u}$ is given by:

$$\mathcal{L}_\alpha(\boldsymbol{u}) \triangleq \|\mathcal{F}_\alpha(Z(\boldsymbol{u}))\|_2^2 = \|(\mathcal{I} - \mathcal{M} \circ \mathcal{T}_\alpha)(Z(\boldsymbol{u}))\|_2^2. \quad (10)$$

Specifically, for each view $i$, we apply $\mathcal{L}_\alpha$ to the feature map $\hat{z}_0^{(i)}(\boldsymbol{u}; t)$ at each point $\boldsymbol{u}$, derived from Eq. (6), for time step $t$. We use one preceding and one subsequent view as reference neighboring views for adjustments. The loss function is formulated as:

$$\mathcal{L}_i(\boldsymbol{u}; t) = \sum_{j \in \mathrm{A}_i} \omega_j \|\hat{z}_0^{(i)}(\boldsymbol{u}; t) - \mathcal{M}^{(j)} \circ \mathcal{T}_\alpha(\hat{z}_0^{(i)}(\boldsymbol{u}; t))\|_2^2, \quad (11)$$

where $A_i = [i - 1, i + 1] \setminus \{i\}$ denotes the set of adjacent views for the $i$-th view, $\{\omega_j\}$ are the weighted coefficients that $\sum \omega_j = 1$, and $\mathcal{M}^{(j)}$ represents the feature interaction block for the $j$-th view. Now, we can calculate the gradient of the loss $\mathcal{L}_i(\boldsymbol{u}; t)$ with respect to $z_t^{(i)}(\boldsymbol{u})$ at each spatial point $\boldsymbol{u}$ across all views to get a gradient map denoted as $\nabla_{z_t} \mathcal{L}_t \in \mathbb{R}^{F \times h \times w \times c}$. We then use the gradient map to rectify the denoising direction (detailed in Appendix B.2) at each time step $t$, formulated as:

$$\hat{\varepsilon}_\theta(z_t, t) \leftarrow \varepsilon_\theta(z_t, t) + s(t) \sqrt{1 - \overline{\alpha}_t} \nabla_{z_t} \mathcal{L}_t, \quad (12)$$

where $s(t)$ is the rectification scale with respect to $t$. Due to the high computational cost and time when calculating the UNet Jacobian $\frac{\partial \varepsilon_\theta}{\partial z_t}$ during gradient computation, we theoretically (detailed in Appendix B.3) and experimentally (detailed in Appendix D) show that neglecting the UNet Jacobian term causes negligible changes in rectification results. The following theorem provides a specific explanation.

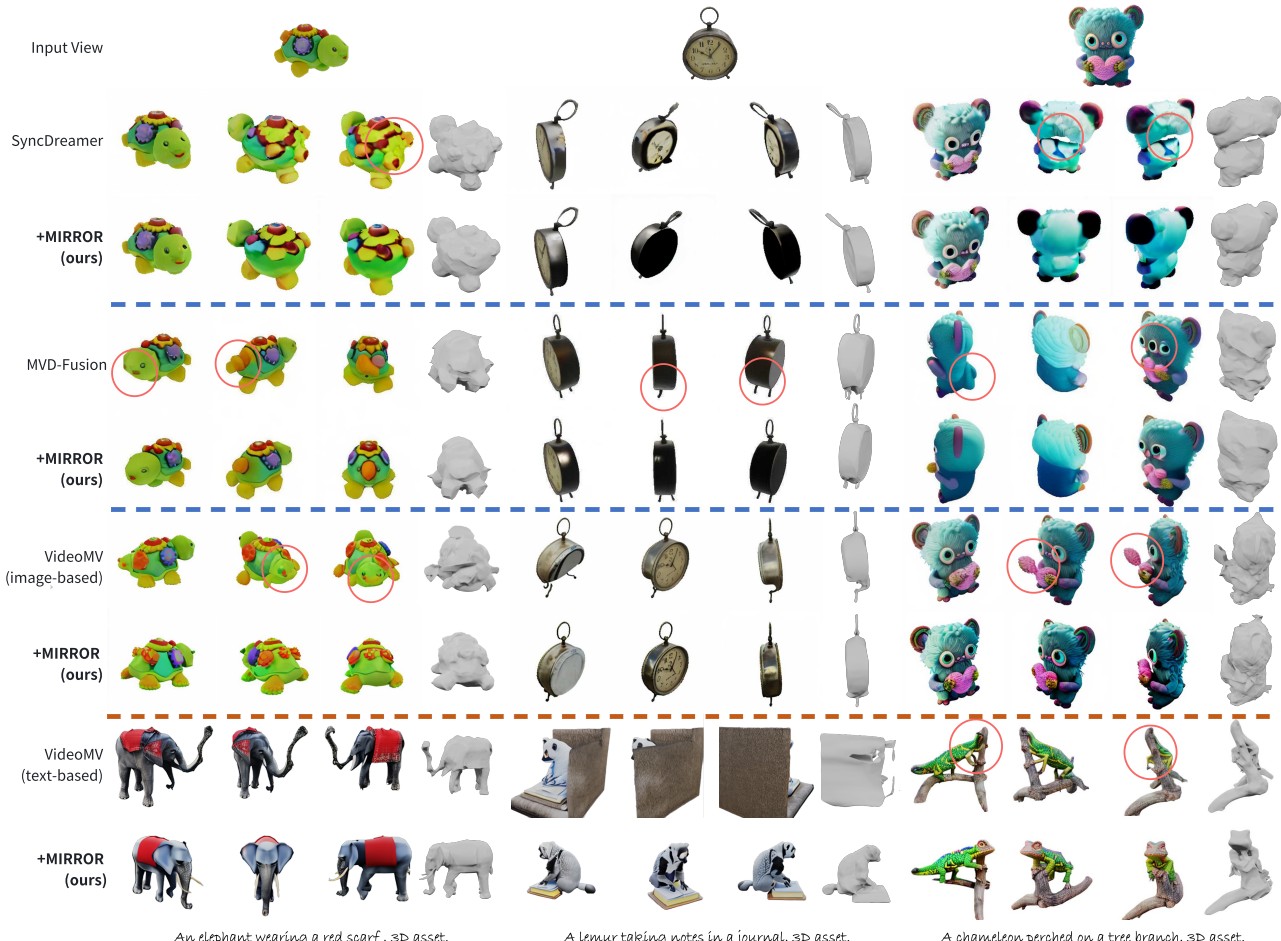

Figure 5: Qualitative Results. The images at the top are multi-view images generated by each baseline, while those at the bottom showcase the consistency rectification achieved through MIRROR. For easier observation, some areas with Janus Problem or content drifting are marked with red circles. For visual consistency, we replace the gray background of the generated images from text prompts with a pristine white one.

**Theorem 4.4.** *In the denoising process, the rectification error induced by neglecting the UNet Jacobian term, has an upper bound controlled by a constant $\gamma$, formulated as*

$$\|\nabla_{z_t} \mathcal{L}_t - \frac{1}{\sqrt{\overline{\alpha}_t}} \nabla_{\hat{z}_0} \mathcal{L}_t\| \leq \gamma, \tag{13}$$

*and this error converges to zero as $t \to 0$.*

Therefore, we can rewrite the rectification formula Eq. (12):

$$\hat{\varepsilon}_\theta(z_t, t) \leftarrow \varepsilon_\theta(z_t, t) + s(t) \frac{\sqrt{1 - \overline{\alpha}_t}}{\sqrt{\overline{\alpha}_t}} \nabla_{\hat{z}_0} \mathcal{L}_t, \tag{14}$$

where $\nabla_{\hat{z}_0} \mathcal{L}_t$ denotes the gradient map with respect to $\hat{z}_0$. We demonstrate that Eq. (14) achieves nearly a twofold speedup compared to Eq. (12) (Tab. 2), enabling faster rectification without sacrificing the quality of the generation. The rectified noise $\hat{\varepsilon}_\theta$ is then subsequently fed into Eq. (2) to carry out the sampling procedure utilized in standard DDIM (Song et al., 2020), but with the rectified noise predictions.

## 5. Experiments

### 5.1. Experimental Setup

**Baselines.** In this section, we conduct experimental evaluations based on image-based and text-based multi-view generation tasks. For image-based tasks, we choose Sync-Dreamer (Liu et al., 2023b), MVD-Fusion (Hu et al., 2024), and VideoMV (Zuo et al., 2024) as our baselines. For text-based tasks, we adopt VideoMV as baseline method. Each method provides a pretrained multi-view model for generating multiple views from a single image or a text prompt.

**Metrics.** Building on these baselines, we quantitatively evaluate performance using metrics including PSNR, SSIM (Wang et al., 2004), and LPIPS (Zhang et al., 2018). We also report Clip-Score for both image-based and text-based generation tasks to evaluate the alignment between generated images and the corresponding input.

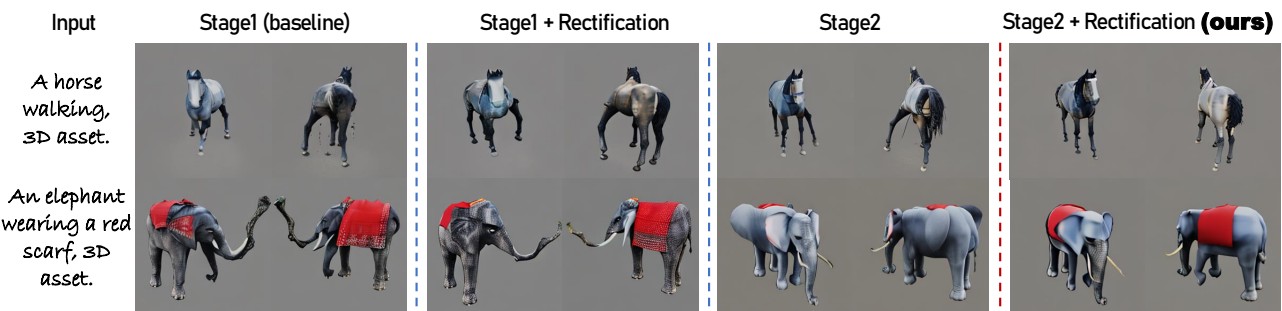

Figure 6: Ablation Study on the Two-Stage Rectification Process. Results generated using baseline methods often exhibit severe Janus Problem. Attempts to rectify these directly at stage 1, as well as outcomes from a two-stage baseline process, have proven inadequate in resolving these issues. In contrast, our method significantly reduces these inconsistencies.

**Implementation Details.** We implement MIRROR on top of the baseline models discussed earlier in this section. This integration of the rectification process occurs exclusively during the inference phase, thereby obviating the need for any supplementary training or fine-tuning. Given that TTM relies on depth guidance, we employ the state-of-the-art monocular depth estimation model, Depth-Anything-V2-Small (Yang et al., 2024), to generate depth information. We also provide an ablation study on alternative depth guidance methods, as detailed in Appendix E, and demonstrate that MIRROR's performance improves with the advancement of depth estimation models. Following baselines, we use NeuS (Wang et al., 2021) for 3D reconstruction. Although calculating the UNet Jacobian in Eq. (12) is time-consuming and significantly increases inference time, our method, as described in Eq. (14), shows that by omitting this term and integrating MIRROR into the baseline models, the inference time is considerably reduced. As shown in Tab. 2, our two-stage sampling process is completed in just a few minutes. Additional experimental details are provided in Appendix C.

### 5.2. Comparison with Baseline Methods

We implement MIRROR across diverse open-source multi-view diffusion models and carry out both quantitative and qualitative comparisons.

**Qualitative Analysis.** For image-based generation, Fig. 5 clearly illustrate that although baseline models can produce plausible views from angles near the input perspective, they exhibit inconsistencies (Janus Problem and content drift) in lateral and rear views, along with undesirable indentations, and misalignments. Fig. 5 shows that text-based generation of VideoMV also suffers from low-quality outcomes with undesirable artifacts and inconsistencies, such as an 'elephant' with multiple trunks and missing legs, and a 'chameleon' without a head. By comparison, incorporating MIRROR into these baselines successfully resolves the prominent artifacts and multi-face issues encountered in baselines, as observed in Fig. 1 and Fig. 5, which not only

Table 1: Quantitative Comparison with Baselines. Image-based and text-based generation results are evaluated on GSO (Downs et al., 2022) and T³Bench (He et al., 2023) datasets, respectively.

| Method | PSNR ↑ | SSIM ↑ | LPIPS ↓ | ClipS ↑ |
|---|---|---|---|---|
| SyncDreamer | 18.75 | 0.739 | 0.198 | 76.60 |
| **+MIRROR (ours)** | **19.37** | **0.794** | **0.172** | **78.28** |
| MVD-Fusion | 18.90 | 0.785 | 0.172 | 68.66 |
| **+MIRROR (ours)** | **20.15** | **0.801** | **0.168** | **71.52** |
| VideoMV (image-based) | 17.64 | 0.755 | 0.197 | 71.77 |
| **+MIRROR (ours)** | **18.26** | **0.793** | **0.176** | **73.30** |
| VideoMV (text-based) | 21.67 | 0.814 | 0.206 | 26.96 |
| **+MIRROR (ours)** | **24.82** | **0.898** | **0.115** | **32.78** |

enhances consistency but also significantly improves quality and geometric fidelity, clearly surpassing all baseline results. More results are provided in Appendix K.

**Quantitative Analysis.** The quantitative results in Tab. 1 are in strong concordance with the visualization outcomes. Incorporating MIRROR significantly boosts 3D consistency-related metrics (PSNR, SSIM, and LPIPS) over all baseline methods, clearly demonstrating notable enhancements in consistency and quality for multi-view generation. More-over, we calculate the average CLIP score between generated images and the input, and we observed a marked increase in it, indicating stronger alignment between the generated content and the input. Moreover, 3D reconstruction metrics are also reported in Appendix C.4.

**Inference Time.** Our experiments are conducted using a single NVIDIA L40S GPU, with the two-stage sampling process completing in just a few minutes. Notably, incorporating MIRROR into the baseline models does not result in a substantial increase in computational load or processing time. We test the inference time of the baseline methods and the methods with MIRROR applied, as shown in Tab. 2. Although the computation of the UNet Jacobian is time-

Table 2: Inference Time of Baseline Models and Their MIRROR-Enhanced Versions. The time multiples relative to the baseline are shown in parentheses.

| Models | Baseline | MIRROR (w. UNet Jacobian) | MIRROR (ours) (w/o UNet Jacobian) |
|---|---|---|---|
| **SyncDreamer** | 16 s | 91 s (×5.7) | 43 s (×2.7) |
| **MVD-Fusion** | 20 s | 112 s (×5.6) | 52 s (×2.6) |
| **VideoMV (image-based)** | 22 s | 103 s (×4.7) | 64 s (×2.9) |
| **VideoMV (text-based)** | 34 s | 154 s (×4.5) | 98 s (×2.9) |

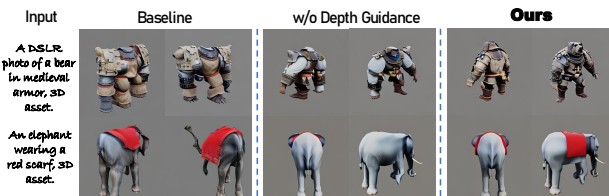

Figure 7: Ablation Study on Depth Guidance. TTM without depth guidance exhibits errors in correspondence, leading to unsatisfactory and inconsistent results.

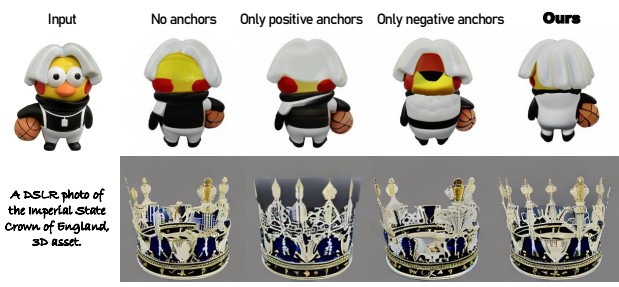

Figure 8: Ablation Study on Dual-Anchor-Based Fusion. Alternative methods yield suboptimal results, while the dual-anchor-based method (ours) markedly reduces background interference and improves the continuity of local features.

consuming (w. UNet Jacobian), increasing the inference time by 4.5 to 5.7 times compared to the baselines, our method demonstrates that the UNet Jacobian term can be omitted, leading to a significant reduction in inference time, with only a 2.6 to 2.9 times increase (w/o UNet Jacobian). This computational cost is still minimal when compared to training-based methods, enabling the generation of high-quality dense images within minutes.

### 5.3. Ablation Study

***Two-Stage Rectification Process.*** Fig. 6 highlights that replacing our two-stage rectification process with a single-stage correction (the second column) fails to resolve the Janus problem evident in baselines, as inaccuracies in depth extraction lead to deformed 'horse' and 'elephant' models with missing or unrealistic features. And relying solely on a two-stage denoising process without rectification (the third column) still leads to inconsistencies. Conversely, MIR-

ROR utilizes depth information from the initial denoising stage for more accurate tracking and rectification in the subsequent stage, which significantly improves multi-view consistency and model fidelity, as shown in the final column.

***Depth Guidance.*** Within TTM, our approach leverages depth information to obtain more accurate spatial correspondences between different viewpoints. However, in the absence of depth guidance (i.e., $z = 0$ in Eq. (4)), it fails to precisely establish correspondences between adjacent views, leading to unsatisfactory results, as depicted in the second column of Fig. 7, with the missing head in the first row and content drift problem of 'red scarf' in the second row.

***Dual-Anchor-Based Fusion.*** To validate our method's effectiveness and generalizability, we conduct ablation studies on multi-view generation against white and gray backgrounds. Our findings, detailed in Fig. 8, show that moving away from a dual-anchor approach to using average pooling or single-anchor feature fusion leads to various issues. Specifically, omitting the anchor-based approach results in excessive saturation and discontinuity in details, while using only positive anchors improve local feature precision but is prone to background interference. Sole reliance on negative anchors minimizes distractions but lacks focus on essential target characteristics. Conversely, our integrated approach, which combines positive and negative anchors, substantially enhances relevant detail attention and consistency, thereby outperforming other methods. More ablation study results are provided in Appendix J.

## 6. Conclusions

In this paper, we introduce MIRROR, a novel plug-and-play technique that effectively resolves inconsistencies in 3D geometry and artifacts in diffusion-based multi-view generation without training. Using trajectory tracking and feature rectification modules, MIRROR successfully enhances the photorealism of synthesized images across views. Generalization evaluations confirm MIRROR's compatibility with various multi-view diffusion models, making it a universal correction tool that markedly improves generation consistency and fidelity, thereby delivering higher-quality multi-view images for 3D asset reconstruction.

## Acknowledgements

This paper is supported by National Key R&D (Research and Development) Program of China (2021YFA1000403), the National Natural Science Foundation of China (Nos. 12431012, U23B2012, 12471308) and Beijing Natural Science Foundation (1254050).

## Impact Statement

This project aims to provide an effective, training-free tool for improving the generative quality of multi-view diffusion models. Our approach enhances the 3D consistency of multi-view images while simultaneously improving the overall quality of the generated content. However, similar to other generative technologies, there exists a potential for misuse. Malicious entities could combine this method with targeted generative models to produce misleading or harmful 3D content. Addressing these ethical considerations is critical, and future research in 3D generation must prioritize understanding and mitigating these risks to ensure the responsible application of such technologies.

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

## A. Appendix Overview

In the appendix, we provide the following supplementary material to support the main text:

- Derivation and Proofs (Appendix B);

- Additional Implementation Details (Appendix C);

- Convergence and Stability Analysis of MIRROR (Appendix D);

- Depth Guidance (Appendix E);

- Discussion with Multi-View Depth Estimation (Appendix F);

- Correspondence Visualization of Trajectory Tracking Transformation (Appendix G);

- Discussion with Point Tracking Method (Appendix H);

- Limitations and Future Work (Appendix I);

- More Ablation Studies (Appendix J);

- More Results (Appendix K);

- Failure Cases (Appendix L).

## B. Derivation and Proofs

### B.1. Proof of Proposition 4.2

*Proof.* Based on Definition 4.1 of TTM, the trajectory tracking error between the predicted image $\hat{x}_0(x_t, t)$ and the true image $x_0$ can be scaled as:

$$\|\mathcal{T}_\alpha(\hat{x}_0(x_t, t)) - \mathcal{T}_\alpha(x_0)\|^2 \leq \|\boldsymbol{u}_\alpha(\hat{x}_0(x_t, t)) - \boldsymbol{u}_\alpha(x_0)\|^2 \leq r^2\|\mathcal{H}(\hat{x}_0(x_t, t)) - \mathcal{H}(x_0)\|^2 \tag{15}$$

Using the first-order Taylor expansion, we obtain:

$$\mathcal{H}(x_0) = \mathcal{H}(\hat{x}_0(x_t, t)) + \nabla\mathcal{H}(x_0)(x_0 - \hat{x}_0(x_t, t)) + o(\|x_0 - \hat{x}_0(x_t, t)\|), \tag{16}$$

where $o$ represents a higher-order infinitesimal. And $\mathcal{H}$ is a pretrained ViT network with a continuous, bounded gradient that outputs scale-consistent absolute depth, showing that the depth-level error is of the same order as the pixel-level error:

$$\|\mathcal{H}(\hat{x}_0(x_t, t)) - \mathcal{H}(x_0)\| \approx \|\nabla\mathcal{H}(x_0)(x_0 - \hat{x}_0(x_t, t))\| \simeq O(\|x_0 - \hat{x}_0(x_t, t)\|). \tag{17}$$

Then, substituting Eq. (17) into Eq. (15) results in the following inequality:

$$\|\mathcal{T}_\alpha(\hat{x}_0(x_t, t)) - \mathcal{T}_\alpha(x_0)\|^2 \leq O(\|\hat{x}_0(x_t, t) - x_0\|^2). \tag{18}$$

Moreover, note that the right side of the inequality Eq. (18) is equivalent to the Kullback-Leibler (KL) divergence between forward process posteriors $q$ and reverse process posteriors estimation $p_\theta$ of diffusion process:

$$O(\|\hat{x}_0(x_t, t) - x_0\|^2) \simeq \text{KL}(q(x_{t-1}|x_t, x_0)\|p_\theta(x_{t-1}|x_t)), \tag{19}$$

since both terms represent the training objective of the diffusion model. Therefore, in the inference process, the trajectory tracking error is controlled by the diffusion model, which has a finite upper bound. Besides, as $t$ decreases, the predicted image $\hat{x}_0(x_t, t)$ approaches the clean image $x_0$, causing the error to converge to zero. □

## B.2. The Derivation of the Rectification Formula Eq. (12)

According to Bayes' Theorem, the objective function can be decomposed into the following two terms:

$$\nabla_{z_t} \log p_\theta(z_t|c) = \nabla_{z_t} \log p_\theta(z_t) + \nabla_{z_t} \log p_\theta(c|z_t), \tag{20}$$

where $c$ denotes the azimuth angles of the cameras in the multi-view images. Furthermore, since $p_\theta$ is a Gaussian distribution, the transformation relationship between the score function and the denoiser of the diffusion model can be easily derived, i.e.,

$$\nabla_{z_t} \log p_\theta(z_t) = -\frac{1}{\sqrt{1-\overline{\alpha}_t}} \varepsilon_\theta(z_t, t). \tag{21}$$

According to Eq. (20) and Eq. (21), we can derive the following expression:

$$\varepsilon_\theta(z_t, t, c) = \varepsilon_\theta(z_t, t) - \sqrt{1-\overline{\alpha}_t} \nabla_{z_t} \log p_\theta(c|z_t). \tag{22}$$

By substituting the exponential form of the loss function, $\exp(-\mathcal{L}_t)$, into Eq. (22), the rectification formula of the denoising direction is derived as follows:

$$\hat{\varepsilon}_\theta(z_t, t, c) \leftarrow \varepsilon_\theta(z_t, t) + s(t) \sqrt{1-\overline{\alpha}_t} \nabla_{z_t} \mathcal{L}_t, \tag{23}$$

where $s(t)$ is the rectification scale with respect to $t$. For brevity, $\hat{\varepsilon}_\theta(z_t, t, c)$ is denoted as $\hat{\varepsilon}_\theta(z_t, t)$ in the main paper.

## B.3. Proof of Theorem 4.4

*Proof.* First, by applying the chain rule, the gradient $\nabla_{z_t} \mathcal{L}_t$ can be expanded as follows:

$$\nabla_{z_t} \mathcal{L}_t = \frac{\partial \hat{z}_0}{\partial z_t} \nabla_{\hat{z}_0} \mathcal{L}_t = \frac{1}{\sqrt{\overline{\alpha}_t}} \left(1 - \sqrt{1-\overline{\alpha}_t} \frac{\partial \varepsilon_\theta}{\partial z_t}\right) \nabla_{\hat{z}_0} \mathcal{L}_t, \tag{24}$$

with $\frac{\partial \varepsilon_\theta}{\partial z_t}$ being the UNet Jacobian of the diffusion model. When the UNet Jacobian term is neglected, the gradient error between the two terms can be expressed as:

$$\left\| \nabla_{z_t} \mathcal{L}_t - \frac{1}{\sqrt{\overline{\alpha}_t}} \nabla_{\hat{z}_0} \mathcal{L}_t \right\| = \frac{\sqrt{1-\overline{\alpha}_t}}{\sqrt{\overline{\alpha}_t}} \left\| \frac{\partial \varepsilon_\theta}{\partial z_t} \nabla_{\hat{z}_0} \mathcal{L}_t \right\|. \tag{25}$$

Note that as $t$ decreases, the noise $\varepsilon_\theta(z_t, t)$ becomes increasingly independent of the data distribution, causing the UNet Jacobian term to converge to zero as $t \to 0$. Then, $\| \frac{\partial \varepsilon_\theta}{\partial z_t} \nabla_{\hat{z}_0} \mathcal{L}_t \| \to 0$ as $t \to 0$. Therefore, there exists a constant $\gamma$ such that $\| \frac{\partial \varepsilon_\theta}{\partial z_t} \nabla_{\hat{z}_0} \mathcal{L}_t \| \leq \gamma$, for all $t$. Moreover, since the coefficient $\frac{\sqrt{1-\overline{\alpha}_t}}{\sqrt{\overline{\alpha}_t}} \in [0, 1]$, the gradient error of Eq. (25) has a finite upper bound.

Furthermore, after individually rectifying the noise estimation based on the two gradients in Eq. (12) and Eq. (14), respectively, the error of $z_{t-1}$ obtained through sampling process also converges to zero as $t$ decreases. The detailed derivation process is as follows:

$$
\begin{aligned}
\left\| z_{t-1}^{(z_t)} - z_{t-1}^{(\hat{z}_0)} \right\| &= \left\| \left( \sqrt{\overline{\alpha}_{t-1}} \left( \frac{z_t - \sqrt{1-\overline{\alpha}_t} \hat{\varepsilon}_\theta^{(z_t)}}{\sqrt{\overline{\alpha}_t}} \right) + \sqrt{1-\overline{\alpha}_{t-1}} \hat{\varepsilon}_\theta^{(z_t)} \right) \right. \\
&\quad \left. - \left( \sqrt{\overline{\alpha}_{t-1}} \left( \frac{z_t - \sqrt{1-\overline{\alpha}_t} \hat{\varepsilon}_\theta^{(\hat{z}_0)}}{\sqrt{\overline{\alpha}_t}} \right) + \sqrt{1-\overline{\alpha}_{t-1}} \hat{\varepsilon}_\theta^{(\hat{z}_0)} \right) \right\| \\
&= \left| \frac{\sqrt{\overline{\alpha}_{t-1}}\sqrt{1-\overline{\alpha}_t}}{\sqrt{\overline{\alpha}_t}} - \sqrt{1-\overline{\alpha}_{t-1}} \right| \cdot \left\| \hat{\varepsilon}_\theta^{(z_t)} - \hat{\varepsilon}_\theta^{(\hat{z}_0)} \right\| \\
&= s(t) \left| \frac{\sqrt{\overline{\alpha}_{t-1}}(1-\overline{\alpha}_t)}{\sqrt{\overline{\alpha}_t}} - \sqrt{(1-\overline{\alpha}_{t-1})(1-\overline{\alpha}_t)} \right| \cdot \left\| \nabla_{z_t} \mathcal{L}_t - \frac{1}{\sqrt{\overline{\alpha}_t}} \nabla_{\hat{z}_0} \mathcal{L}_t \right\| \\
&= s(t) \frac{\sqrt{1-\overline{\alpha}_t}}{\sqrt{\overline{\alpha}_t}} \left| \frac{\sqrt{\overline{\alpha}_{t-1}}(1-\overline{\alpha}_t)}{\sqrt{\overline{\alpha}_t}} - \sqrt{(1-\overline{\alpha}_{t-1})(1-\overline{\alpha}_t)} \right| \cdot \left\| \frac{\partial \varepsilon_\theta}{\partial z_t} \nabla_{\hat{z}_0} \mathcal{L}_t \right\| \\
&\leq \gamma \cdot \mathcal{A}(t), 
\end{aligned}
\tag{26}
$$

where the superscripts $(z_t)$ and $(\hat{z}_0)$ correspond to the two gradient correction methods in Eq. (12) and Eq. (14), respectively. And the coefficient $\mathcal{A}(t) = s(t) \frac{\sqrt{1-\overline{\alpha}_t}}{\sqrt{\overline{\alpha}_t}} \left| \frac{\sqrt{\overline{\alpha}_{t-1}}(1-\overline{\alpha}_t)}{\sqrt{\overline{\alpha}_t}} - \sqrt{(1-\overline{\alpha}_{t-1})(1-\overline{\alpha}_t)} \right| \to 0$ as time step $t \to 0$. $\qquad \square$

---

**Algorithm 1** Framework of MIRROR

---

**Input**: a single image or a text prompt $c$
**Pretrained baseline model**: $\mathcal{BM}$
**Parameter**: rectification scale $s(t)$ with respect to $t$
**Output**: multi-view consistent images $\hat{x}_0$

1: $z_T \leftarrow$ sample from $\mathcal{N}(0, \boldsymbol{I})$
2: $z_t \leftarrow z_T$.
3: **for** $t = T, T-1, \cdots, 1$ **do**
4:    $\varepsilon_\theta \leftarrow \mathcal{BM}(z_t, c, t)$.
5:    $z_{t-1} \leftarrow DDIM(z_t, \varepsilon_\theta)$
6: **end for**
7: $\hat{x}_0 \leftarrow \mathcal{D}(z_0)$.
8: $\boldsymbol{\mathcal{Z}}'_{absolute} \leftarrow DEPTH\_ESTIMATION(\hat{x}_0)$.
9: $z_T \leftarrow DDIM\_INVERSION(z_0, \mathcal{BM}, T, c)$.
10: $z_t \leftarrow z_T$.
11: **for** $t = T, T-1, \cdots, 1$ **do**
12:    $\varepsilon_\theta \leftarrow \mathcal{BM}(z_t, c, t)$.
13:    $\mathcal{L}_t \leftarrow \|\mathcal{F}_\alpha(z_t, \boldsymbol{\mathcal{Z}}'_{absolute})\|_2^2$.
14:    $\hat{\varepsilon}_\theta \leftarrow \varepsilon_\theta + s(t) \frac{\sqrt{1-\overline{\alpha}_t}}{\sqrt{\overline{\alpha}_t}} \nabla_{\hat{z}_0} \mathcal{L}_t$.
15:    $z_{t-1} \leftarrow DDIM(z_t, \hat{\varepsilon}_\theta)$
16: **end for**
17: $\hat{x}_0 \leftarrow \mathcal{D}(z_0)$
18: **return** $\hat{x}_0$

---

### B.4. Group Homomorphism Property of Operator $\mathcal{M} \circ \mathcal{T}_\alpha$

To prove this property, we first establish the following lemma.

**Lemma B.1.** *The trajectory tracking operator $\mathcal{T}_\alpha$ is a group homomorphism. Specifically, for any $\alpha_1, \alpha_2 \in \mathbb{R}$, it holds that:*
$\mathcal{T}_{\alpha_1} \circ \mathcal{T}_{\alpha_2} = \mathcal{T}_{\alpha_1 + \alpha_2}$.

*Proof.* It is important to note that $\mathcal{T}_\alpha$ represents a rotation operator of angle $\alpha$ around the $y$-axis within the special orthogonal group $SO(3)$, which can be expressed using the matrix exponential of the Lie algebra element associated with the rotation group. Specifically, it is given by:

$$\mathcal{T}_\alpha = \exp(\alpha A), \ A = \begin{pmatrix} 0 & 0 & -1 \\ 0 & 0 & 0 \\ 1 & 0 & 0 \end{pmatrix}. \tag{27}$$

By leveraging the properties of the matrix exponential, we observe that the composition of two such rotation operators results in a new rotation operator. That is, the combination of two rotations is given by:

$$\mathcal{T}_{\alpha_1} \circ \mathcal{T}_{\alpha_2} = \exp(\alpha_1 A) \cdot \exp(\alpha_2 A) = \exp((\alpha_1 + \alpha_2)A) = \mathcal{T}_{\alpha_1 + \alpha_2}. \tag{28}$$

Therefore, the rotation operators $\mathcal{T}_\alpha$ are closed under the operation of parameter addition, demonstrating the group property of $SO(3)$ with respect to the rotation angle. $\square$

**Proposition B.2.** *For any $\alpha_1, \alpha_2 \in \mathbb{R}$, the composition of two operators $\mathcal{M} \circ \mathcal{T}_{\alpha_1}$ and $\mathcal{M} \circ \mathcal{T}_{\alpha_2}$ adheres to the following operational relation:*

$$(\mathcal{M} \circ \mathcal{T}_{\alpha_1}) \circ (\mathcal{M} \circ \mathcal{T}_{\alpha_2}) = \mathcal{M} \circ \mathcal{T}_{\alpha_1 + \alpha_2}. \tag{29}$$

*In other words, $\mathcal{M} \circ \mathcal{T}_\alpha$ is a group homomorphism.*

Now, we proceed to prove Proposition B.2.

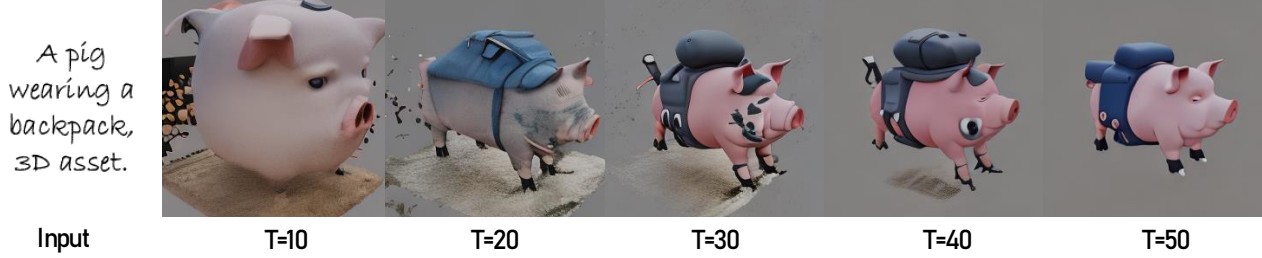

Figure 9: Ablation Study on the Step $T$ in DDIM Inversion. When $T$ is too small, noise disrupts the sampling; too large, it increases inference time. Therefore, after balancing inference time and generation quality, we set $T = 50$ .

*Proof.* It is important to observe that, since the operator $\mathcal{T}_\alpha$ is solely dependent on the coordinates of the spatial points and is independent of the feature values, it can be deduced that

$$(\mathcal{M} \circ \mathcal{T}_{\alpha_1}) \circ (\mathcal{M} \circ \mathcal{T}_{\alpha_2}) = \mathcal{M} \circ (\mathcal{T}_{\alpha_1} \circ \mathcal{T}_{\alpha_2}) = \mathcal{M} \circ \mathcal{T}_{\alpha_1 + \alpha_2}, \tag{30}$$

where the second equality is derived from Lemma B.1. □

## C. Additional Implementation Details

### C.1. Feature Rectification Module

In Algorithm 1, we systematically present the pseudocode for the entire rectification algorithm. Below are the parameter selection strategies for several key parameters used in FRM. **First**, we have determined the weights for the fusion of positive and negative anchors at $(m^+ = 0.5,\ m^- = 0.5)$ based on empirical observations. Fig. 8 shows the $(m^+ = 1,\ m^- = 0)$ would cause grayish or bright tones and $(m^+ = 0,\ m^- = 1)$ may fail to focus on the current point, leading to poor results. To maintain general applicability, we chose $(m^+ = 0.5,\ m^- = 0.5)$ as the robust option. **Second**, we define the rectification scale $s(t)$ to match the consistency gradient magnitude with $\varepsilon_\theta$ of each baseline. As geometric prototypes primarily develop during the initial sampling stages of the inference process, with subsequent stages dedicated to refining detailed textures, it is essential to focus on the early phases since we aim to adjust geometric shapes to address inconsistencies such as Janus Problem. In the later stages, we intentionally decrease the correction scale to prevent overcorrection. Besides, based on experimental observations, the first five denoising steps contain excessive noise, and applying the correction algorithm during this stage leads to redundant information. Therefore, we start applying our correction algorithm after the fifth step. Technically, we define the rectification scale $s(t)$ as a gradient function dependent on $t$. Its specific form is given by Eq. (31) for SyncDreamer and MVD-Fusion, and by Eq. (32) for VideoMV, and can be further adapted to accommodate different models.

$$s(t) = \begin{cases} 0.002, & t \in (40, 45], \\ 0.008, & t \in (30, 40], \\ 0.005, & t \in (20, 30], \\ 0.003, & t \in (10, 20], \\ 0.001, & t \in [1, 10]. \end{cases} \tag{31} \qquad s(t) = \begin{cases} 0.005, & t \in (40, 45], \\ 0.024, & t \in (30, 40], \\ 0.015, & t \in (20, 30], \\ 0.008, & t \in (10, 20], \\ 0.003, & t \in [1, 10]. \end{cases} \tag{32}$$

Furthermore, adjustments per model are feasible. **Third**, the weights $\{\omega_j\}$ in the rectification formulation of the adjacent views, both preceding and succeeding, are assigned randomly, as the feature information provided by these two neighboring views is considered equally important. **Additionally**, we acknowledge that camera parameter noise is unavoidable, but its impact is minimal since the baselines were trained with standard parameters. To alleviate this, we employed block-based trajectory tracking, using a $3 \times 3$ block to fuse the corresponding features. While we experimented with larger windows, it introduced excessive redundant information, ultimately affecting performance. Additionally, increasing the window size further increased the computational load. As a trade-off between computation time and generation quality, we selected the $3 \times 3$ block size.

## C.2. The Step of DDIM Inversion

The number of DDIM sampling and inversion steps is both set to $T = 50$. The number of DDIM sampling steps in the two-stage process is kept the same as the baselines. And we have tried with different $T$ for DDIM Inversion. As shown in Fig. 9, when $T$ is too small, the added noise becomes excessively coarse, negatively impacting the denoising process in the second stage. Conversely, when $T$ is too large, it significantly increases the overall inference time. Therefore, we have settled on $T = 50$, which strikes a balance between maintaining high quality and ensuring computational efficiency, consistent with the inference process steps.

## C.3. Evaluation Datasets

For image-based generation, we evaluated 100 objects from Google Scanned Objects (GSO) (Downs et al., 2022) dataset, using the 16, 16, and 24 views provided by SyncDreamer, MVD-Fusion, and VideoMV, respectively. Following text-based generation in VideoMV, we used 100 single-object prompts from $T^3$Bench (He et al., 2023), sampling half of the equidistant views (12 views) for neural field reconstruction, employing them as pseudo ground truth for quantitative comparisons.

## C.4. 3D Reconstruction

We follow the baseline SyncDreamer (Liu et al., 2023b) to utilize NeuS (Wang et al., 2021) for 3D reconstruction. Specifically, the training process for one object consists of 10,000 steps with a warm-up of 100 steps and loss weights $\lambda_{rgb} = 0.5$, $\lambda_{mask} = 1.0$, $\lambda_{eikonal} = 0.1$. We provide the commonly used Chamfer Distance (CD) and Volume IoU between ground-truth shapes and reconstructed shapes in Tab. 3 to further validate the effectiveness of our method in improving the quality and consistency of both multi-view images and 3D reconstruction.

Table 3: Quantitative Comparison with Baselines on 3D Reconstruction Metrics. Image-based and text-based generation results are evaluated on GSO and $T^3$Bench datasets, respectively.

| Method | Chamfer Distance ↓ | Volume IoU ↑ |
|---|---|---|
| SyncDreamer | 0.042 | 0.514 |
| **+*MIRROR* (ours)** | **0.039** | **0.530** |
| MVD-Fusion | 0.033 | 0.602 |
| **+*MIRROR* (ours)** | **0.029** | **0.656** |
| VideoMV (image-based) | 0.031 | 0.643 |
| **+*MIRROR* (ours)** | **0.026** | **0.667** |
| VideoMV (text-based) | 0.046 | 0.538 |
| **+*MIRROR* (ours)** | **0.028** | **0.626** |

# D. Convergence and Stability Analysis of MIRROR

We experimentally validate the convergence and stability of our algorithm on baseline methods. Firstly, as shown in Fig. 10 (a), we observe that as $t$ decreases, the predicted noise $\hat{\varepsilon}_\theta$ rectified by MIRROR steadily decreases. Second, Fig. 10 (b) and (c) demonstrates that both the loss gradient error of Eq. (25) and the error about $z_{t-1}$ during one-step DDIM sampling of Eq. (26), which are caused by neglecting the UNet Jacobian term, gradually converge to zero as $t$ decreases. Furthermore, Fig. 11 demonstrates that both rectification formulas effectively produce consistent multi-view images. Therefore, by combining the results in Fig. 10 and Fig. 11, we can conclude that the error introduced by omitting the UNet Jacobian term is negligible. Therefore, using Eq. (14) as the rectification formula is a more efficient and reasonable strategy.

# E. Depth Guidance

We evaluated the performance of the current state-of-the-art monocular depth estimation method, Depth-Anything-V2 (Yang et al., 2024) and an alternative method, MiDaS (Ranftl et al., 2020). Experimental results demonstrate that Depth-Anything-V2 is the optimal choice for depth estimation in the TTM framework.

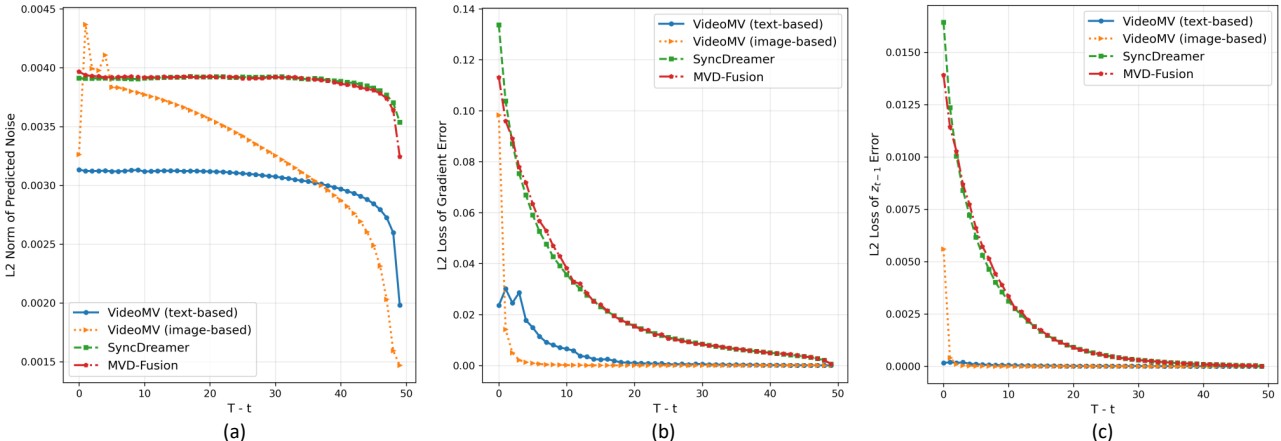

Figure 10: Convergence and Stability Analysis of MIRROR. (a) The trend of predicted noise $\hat{\varepsilon}_\theta$ variation during the inference process with MIRROR; (b) The convergence of gradient errors in Eq. (25); (c) The convergence of one-step DDIM sampling errors in Eq. (26) when omitting the UNet Jacobian term during the inference process.

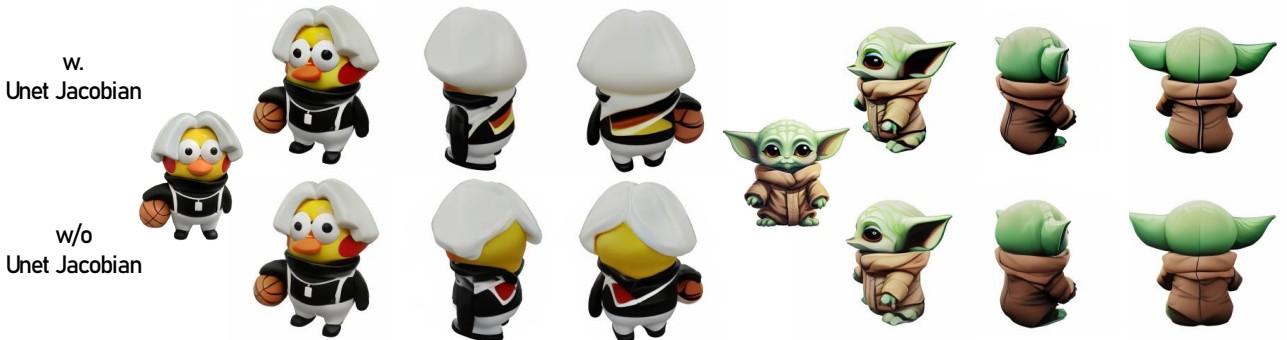

Figure 11: Comparison of generated results with (w.) and without (w/o) considering the UNet Jacobian term, generated from VideoMV (Zuo et al., 2024) rectified with MIRROR. Both rectification formulas are capable of generating multi-view images with strong consistency.

First, we evaluate the error of Eq. (5) for depth estimation in TTM using three image-based baseline models, VideoMV (Zuo et al., 2024), SyncDreamer (Liu et al., 2023b) and MVD-Fusion (Hu et al., 2024), with Depth-Anything-V2 on GSO dataset. Fig. 12 (a) demonstrates that during the denoising process, the trajectory tracking error is minimal with the depth guidance from Depth-Anything-V2 (DA2), and as $t$ decreases from $T$ to 0, it gradually converges to zero. Besides, Fig. 12 (b)-(d) show the comparison of depth estimation errors for each baseline model using DA2 (blue) or MiDaS (orange). In particular, in Fig. 12 (d), for MVD-Fusion, we also utilize the depth values predicted by its pretrained model to facilitate TTM, which serves as the reference standard (green). It is noteworthy that the depth estimation error of DA2 is the smallest among all the baseline methods.

Moreover, we also evaluate the qualitative results (Fig. 13) and quantitative metrics (Tab. 4) on VideoMV (Zuo et al., 2024) rectified with MIRROR. using different monocular depth estimation models for guidance. We found that both the absence of depth estimation and the use of the alternative MiDaS model introduced significant errors to TTM, resulting in 3D inconsistency in the generated multi-view images. In contrast, when using DA2 for depth estimation, the generated results were more consistent and achieved the best performance across all quantitative metrics.

## F. Discussion with Multi-View Depth Estimation

We replaced Depth-Anything-V2 (DA2) with the multi-view depth estimator, DUSt3R (Wang et al., 2024a). As shown in Tab. 5, DUSt3R does not significantly increase inference time but requires more memory, whereas DA2 incurs minimal overhead. In low-memory environments (e.g., a single NVIDIA 3090 GPU), DUSt3R fails to run. While DUSt3R slightly

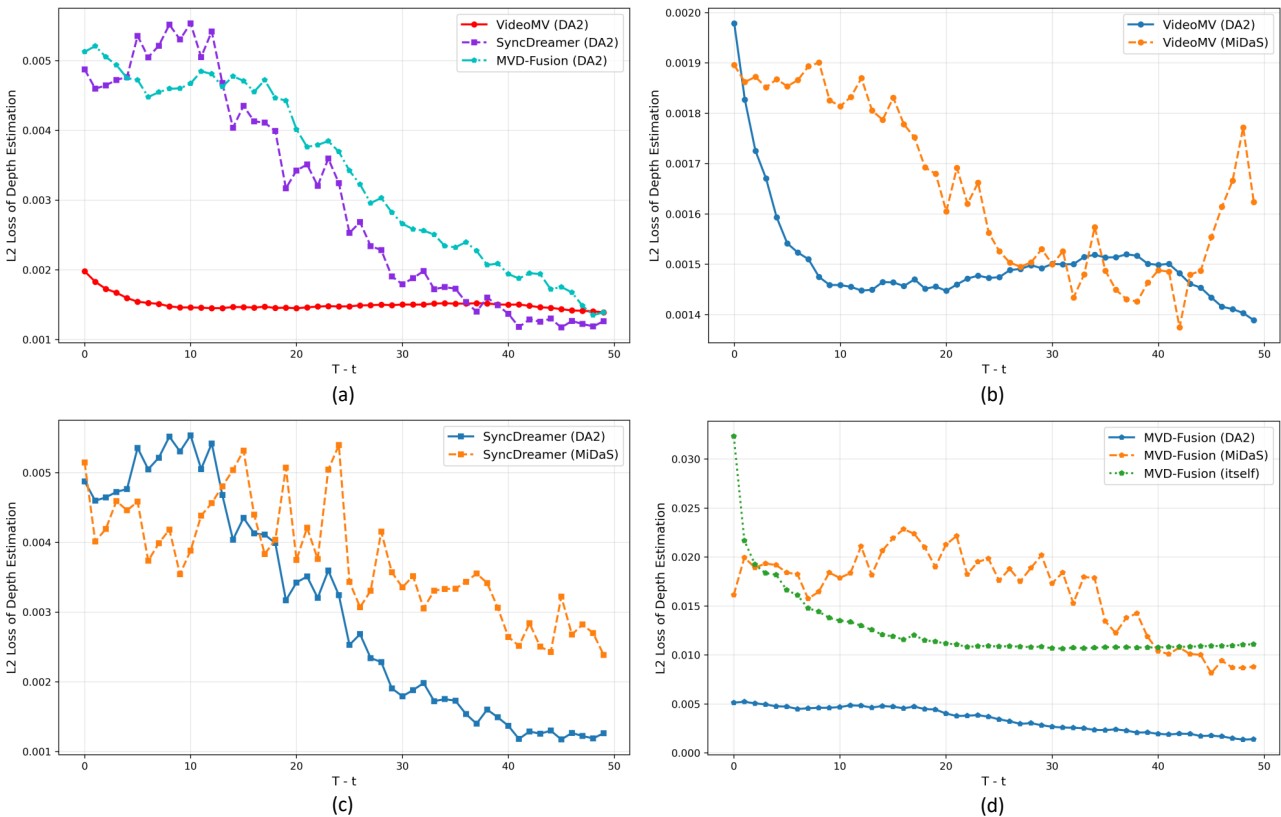

Figure 12: Convergence Analysis of Depth Guidance Models. (a) shows that the trajectory tracking error is minimal with Depth-Anything-V2 (DA2) guidance, converging to zero as $t$ decreases. (b)-(d) compare the depth estimation errors of each baseline model using Depth-Anything-V2 (DA2, blue) and MiDaS (orange). In (d), for MVD-Fusion, we additionally use the depth values predicted by its pretrained model to facilitate TTM, which is set as the reference standard (green).

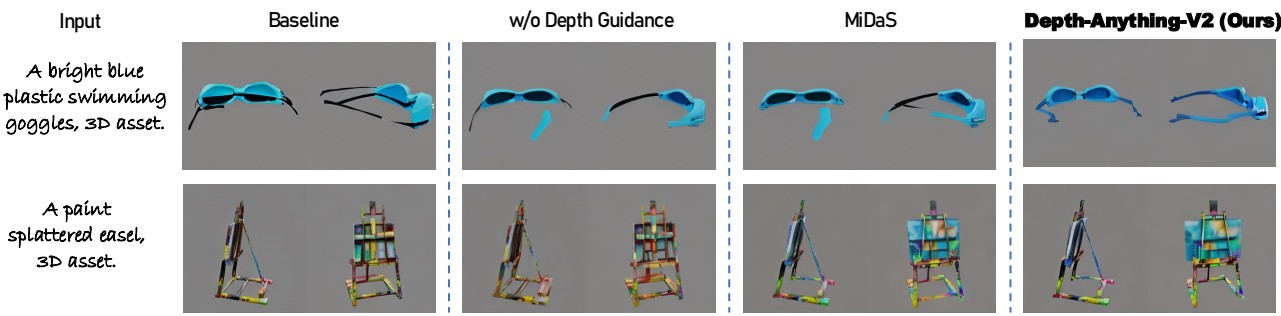

Figure 13: Ablation Study on Depth Guidance Models: MiDaS vs. Depth-Anything-V2.

improves PSNR and LPIPS, SSIM and CLIP Score remain comparable to DA2. Visual results in Fig. 14 show marginal improvement, but in some cases, DUSt3R underperforms DA2.

In summary, multi-view depth provides modest gains but with memory trade-offs. As depth estimation is independent of our core contribution, these results—along with those in Appendix E—show that MIRROR can benefit from ongoing advances in depth estimation.

Table 4: Quantitative Ablation Study on Different Depth Guidance Methods.

| Ablation Study | PSNR ↑ | SSIM ↑ | LPIPS ↓ | ClipS ↑ |
|---|---|---|---|---|
| Baseline | 21.67 | 0.814 | 0.206 | 26.96 |
| w/o Depth Guidance | 16.13 | 0.757 | 0.369 | 26.51 |
| MiDaS | 22.41 | 0.863 | 0.258 | 25.79 |
| **Depth-Anything-V2 (ours)** | **24.82** | **0.898** | **0.115** | **32.78** |

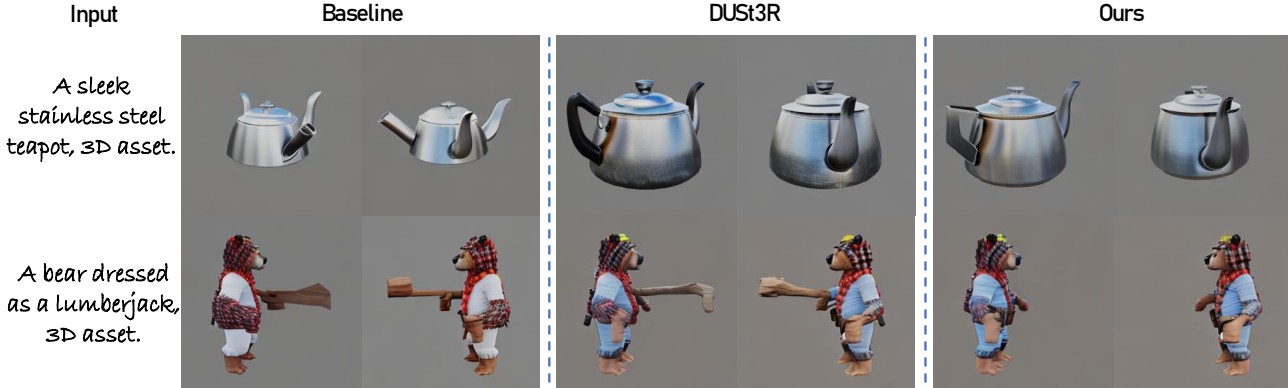

Figure 14: Qualitative Comparison with multi-view depth estimation model DUSt3R (Wang et al., 2024a). The first row shows that depth estimated with DUSt3R leads to more realistic generation results, while our DA2-based approach also produces consistent and reasonable outputs. The second row presents a failure case of DUSt3R-based correction, where our method successfully resolves the hand inconsistency in the baseline. We use the text-based model VideoMV for evaluation, with details consistent with Appendix C.3.

## G. Correspondence Visualization of Trajectory Tracking Transformation

In both image-based and text-based generation tasks, we employ the TTM method to track the correspondence of 3D physical points across adjacent views, as illustrated in Fig. 15. TTM without depth guidance ($z = 0$ in Eq. (4)) exhibits substantial correspondence errors. In contrast, the depth-guided TTM method achieves more precise correspondences, thereby enhancing the outcomes of multi-view generation tasks.

## H. Discussion with the Point Tracking Method

We incorporate the recent point tracking SOTA method, CoTracker3 (Karaev et al., 2024), as a substitute for TTM for comparison. Tab. 6 shows that while CoTracker3 improves the baseline's generation quality to some extent, the gains are less significant than those achieved with TTM. Moreover, Fig. 16 shows CoTracker3 fails to resolve multi-face artifacts, exhibiting noticeable inconsistencies such as multiple legs and misaligned heads, which TTM effectively mitigates. Both quantitative and qualitative results demonstrate that TTM outperforms point tracking methods like CoTracker3 in addressing multi-view inconsistency.

Table 5: Quantitative Comparison with multi-view depth estimation model DUSt3R (Wang et al., 2024a). Our experiments are conducted using a single NVIDIA L40S GPU, with VideoMV (Zuo et al., 2024) serving as the baseline. There is a trade-off between memory usage and performance between the two methods.

| Ablation Study | Time ↓ | Memory Usage ↓ | PSNR ↑ | SSIM ↑ | LPIPS ↓ | ClipS ↑ |
|---|---|---|---|---|---|---|
| Baseline | 34 s | 25.00 GB | 21.67 | 0.814 | 0.206 | 26.96 |
| DUSt3R | 107 s (×3.15) | 27.04 GB (+2.04) | **25.09** | 0.865 | **0.109** | 30.82 |
| DA2 (ours) | **98 s (×2.89)** | **25.08 GB (+0.08)** | 24.82 | **0.898** | 0.115 | **32.78** |

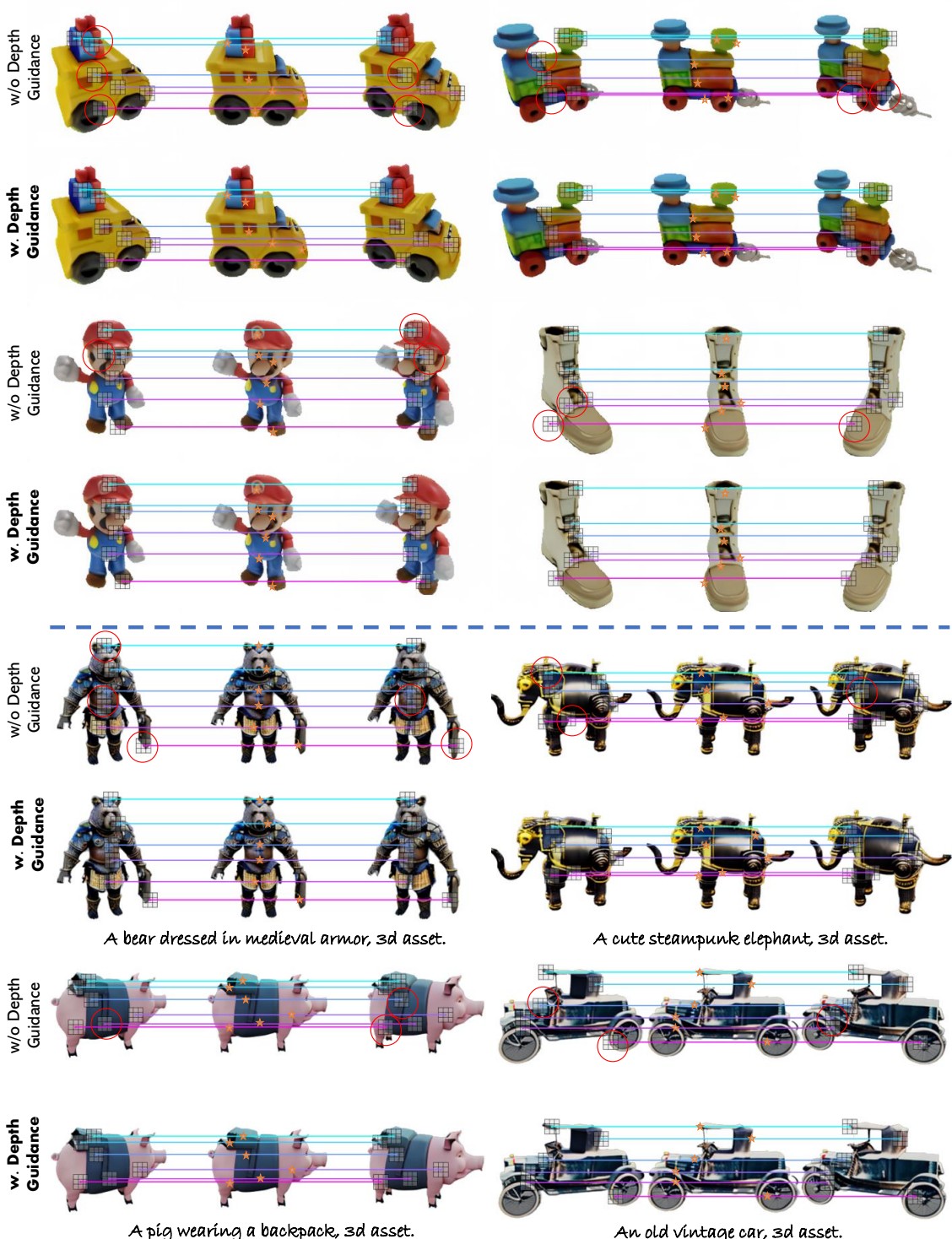

Figure 15: Correspondence visualization of TTM on both image-based and text-based generation. The upper rows illustrate TTM without depth guidance, whereas the lower rows represent the depth-guided TTM. The depth-guided approach more precisely identifies the correspondence of spatial points between adjacent views. For greater accuracy, readers are encouraged to zoom in on the picture. The areas with the Janus Problem or content drifting are marked with red circles.

Table 6: Quantitative Comparison of TTM and the point tracking method, CoTracker3 (Karaev et al., 2024). Notably, while CoTracker3 provides slight improvements over the baseline across all metrics, our method achieves more significant gains.

| Ablation Study | PSNR ↑ | SSIM ↑ | LPIPS ↓ | ClipS ↑ |
|---|---|---|---|---|
| Baseline | 21.67 | 0.814 | 0.206 | 26.96 |
| CoTracker3 | 23.01 | 0.871 | 0.124 | 28.35 |
| **TTM (ours)** | **24.82** | **0.898** | **0.115** | **32.78** |

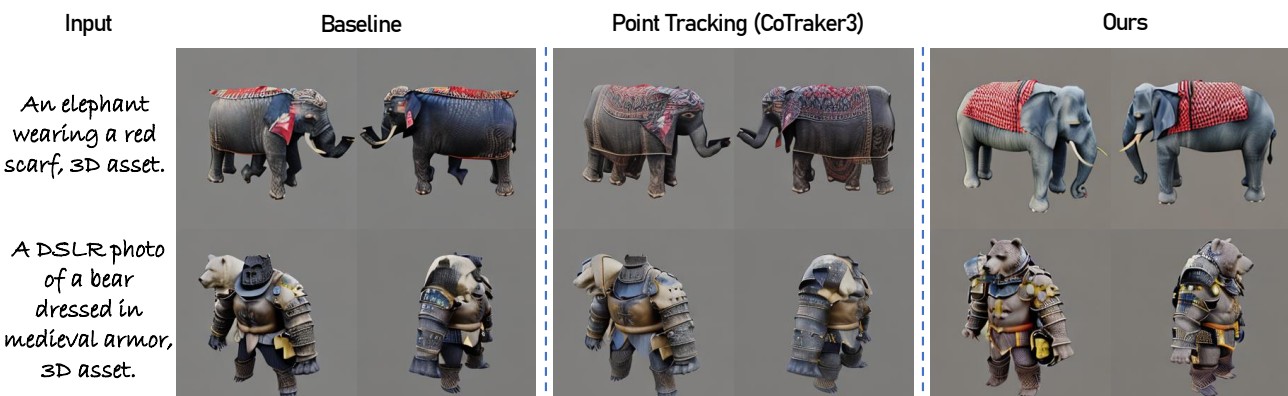

Figure 16: Qualitative Comparison of TTM and the point tracking method, CoTracker3 (Karaev et al., 2024). We use the text-based model VideoMV (Zuo et al., 2024) for evaluation, with details consistent with Appendix C.3. The second column, based on CoTracker3, exhibits noticeable inconsistencies such as multiple legs and misaligned heads. In contrast, our method effectively corrects these issues.

## I. Limitations and Future Work

To ensure uniform multi-view coverage, all baseline models in our paper adopt a fixed elevation setting, and MIRROR is similarly designed under this constraint. While this facilitates consistent geometric modeling and simplifies the correspondence estimation across views, it inevitably limits the applicability of the method in scenarios requiring images from diverse or freely varying viewpoints. This constraint becomes more pronounced in real-world applications where arbitrary camera poses are often necessary. To address this, future work will explore integrating more powerful priors from recent advances in free-viewpoint multi-view diffusion models, thereby extending our trajectory tracking approach to support flexible viewpoint transformations and enhancing the generality of the method.

In addition, our current task setting focuses exclusively on object-level generation. Although this scope allows for fine-grained consistency and controllability, it leaves out a broader class of problems centered around scene-level 3D generation—an increasingly important direction in the field. Nevertheless, the core principles of MIRROR remain relevant. We believe our approach provides valuable insights that can help bridge the gap toward scalable scene-level generation. To enable MIRROR to handle such tasks, future work could incorporate layout-aware priors or explicit scene maps to better account for larger spatial contexts and inter-object relationships.

Ultimately, we view MIRROR as a foundational step toward training-free rectification methods for more complex tasks such as the generation of real-world environments, dynamic scenes, and deformable objects, where geometric consistency and view correspondence remain essential yet challenging components.

## J. More Ablation Studies

We conducted quantitative ablation studies on stages and dual anchors for the text-based generation task (VideoMV) using 50 prompts from T$^3$Bench, comparing our method with alternative approaches, as shown in Tab. 7 and Tab. 8. In Tab. 7, the first three rows correspond to ablation experiments on the two-stage strategy, while the last row shows the metrics for

the results generated using our two-stage rectification. And in Tab. 8, the first three rows are ablation experiments on the dual-anchor points, and the last row shows the metrics of our dual anchors design. The results of both tables demonstrate that our method outperforms others across multiple metrics.

Table 7: Ablation Study Results on Stages.

| Ablation Study | PSNR ↑ | SSIM ↑ | LPIPS ↓ | ClipS ↑ |
|---|---|---|---|---|
| Stage 1 w/o Rec | 21.67 | 0.814 | 0.206 | 26.96 |
| Stage 1 w. Rec | 12.03 | 0.668 | 0.469 | 23.35 |
| Stage 2 w/o Rec | 23.90 | 0.884 | 0.117 | 24.77 |
| **Stage 2 w. Rec (ours)** | **24.82** | **0.898** | **0.115** | **32.78** |

Table 8: Ablation Study Results on Dual Anchors.

| Ablation Study | PSNR ↑ | SSIM ↑ | LPIPS ↓ | ClipS ↑ |
|---|---|---|---|---|
| No Anchors | 24.56 | 0.782 | 0.165 | 27.36 |
| Only Positive Anchors | 23.27 | 0.877 | 0.140 | 28.15 |
| Only Negative Anchors | 24.13 | 0.854 | 0.122 | 28.69 |
| **Dual Anchors (ours)** | **24.82** | **0.898** | **0.115** | **32.78** |

## K. More Results

We present additional results demonstrating the application of MIRROR to rectify baseline models. SyncDreamer (Liu et al., 2023b) and MVD-Fusion (Hu et al., 2024) can generate images from 16 distinct perspectives simultaneously, whereas VideoMV (Zuo et al., 2024) is capable of producing 24 views in a single session. The reduced angle between adjacent frames of VideoMV facilitates a more precise identification of view correspondences, thereby enhancing the accuracy of the refinements. For image-based and text-based generation tasks, please refer to Fig. 17-19 and Fig. 20, respectively. Fig. 17 illustrates the generated results with SyncDreamer as the baseline. In the first two examples, the baseline model exhibits significant issues with content drift. And the subsequent two examples display problematic geometric shapes. Fig. 18 displays the outcomes generated with MVD-Fusion as the baseline. All cases suffer from severe multi-view inconsistency in the baseline. In Fig. 19, the first example exhibits the color drift issue on the 'train' of the baseline model. In the 'teapot' and 'lunch bag', the results generated by the baseline model show inconsistencies across multiple views in terms of shape. And in the last example, it exhibits issues with multiple faces. Fig. 20, which utilizes VideoMV as the baseline model, presents generated results based on text prompts that similarly struggle with a pronounced Janus Problem and low fidelity. After refinement with ***MIRROR***, the generated results significantly surpass all the baseline models. These figures illustrate that our method markedly improves multi-view consistency through incremental refinements during the sampling process, resulting in enhanced quality and superior accuracy.

## L. Failure Cases

Some failure cases are presented in Fig. 21. When the baseline model fails to generate a reasonable geometric structure, is severely flawed, or is completely inconsistent with the prompt, our method struggles to correct such fundamental errors, as the expert priors we rely on are inaccurate. However, by changing the seed to generate well-formed geometric structures from the baseline, this limitation can be mitigated to the greatest extent.

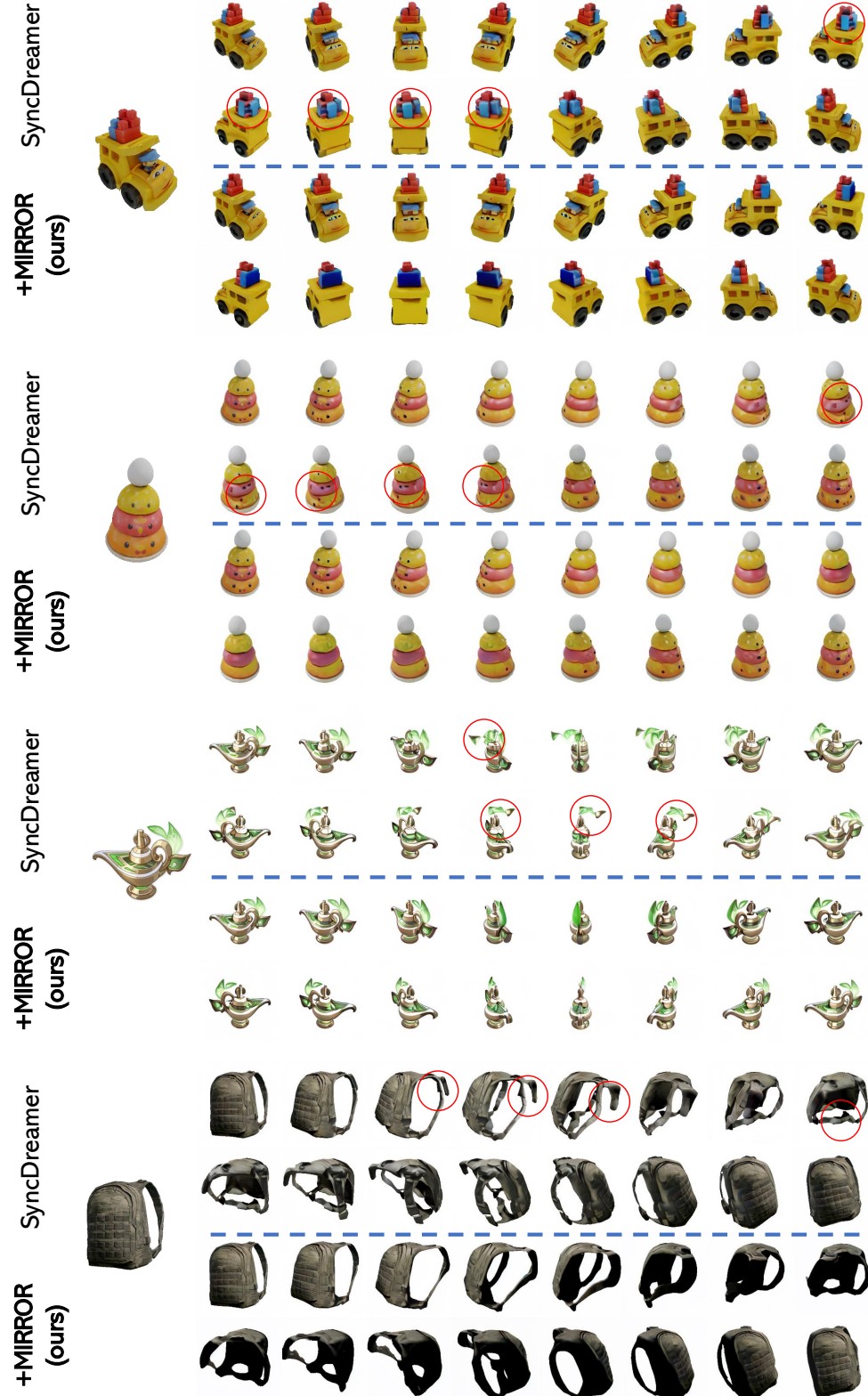

Figure 17: Generated multi-view images by applying MIRROR on SyncDreamer for image-based diffusion rectification. The areas with the Janus Problem or content drifting are marked with red circles.

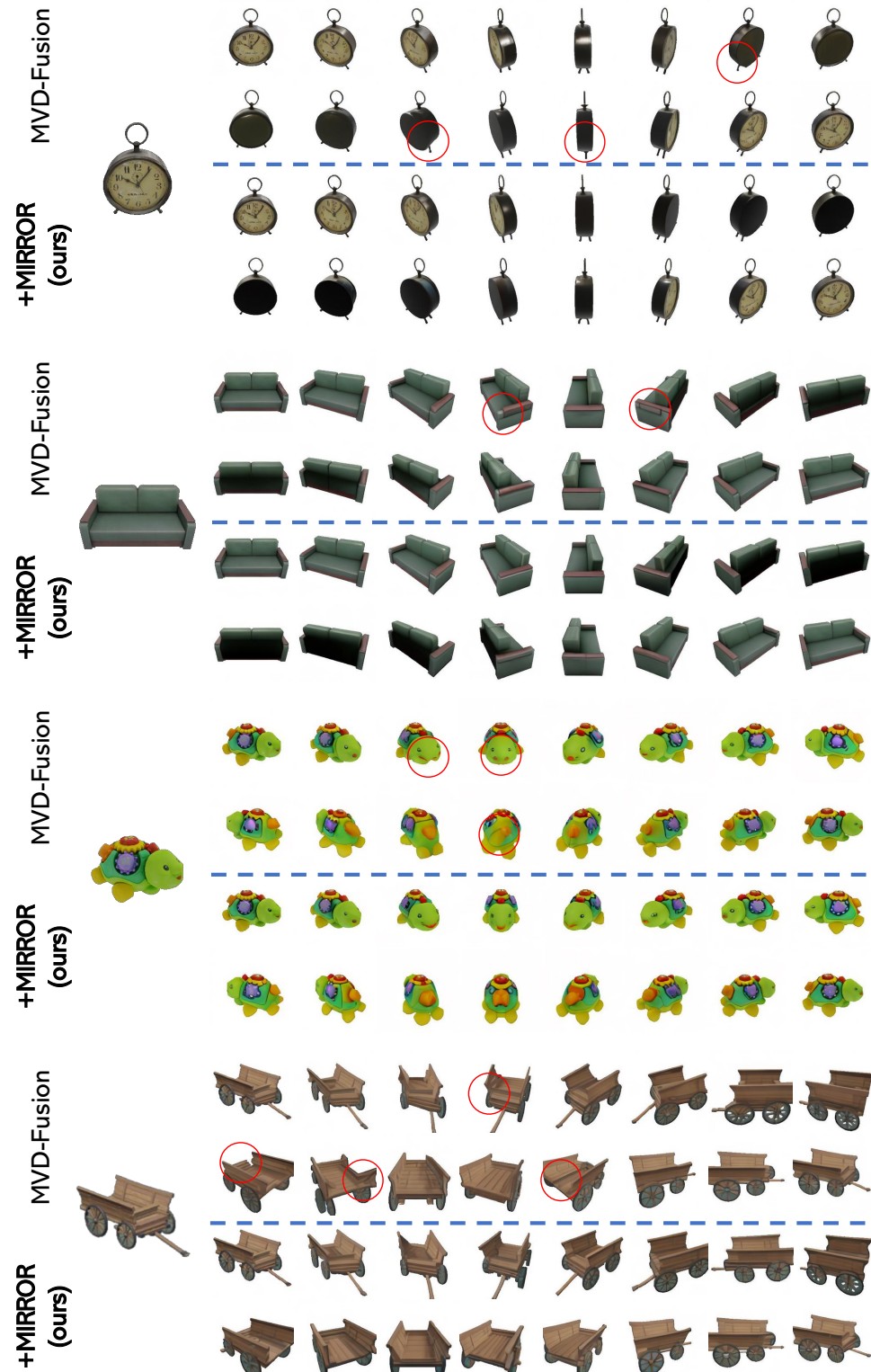

Figure 18: Generated multi-view images by applying MIRROR on MVD-Fusion for image-based diffusion rectification. The areas with the Janus Problem or content drifting are marked with red circles.

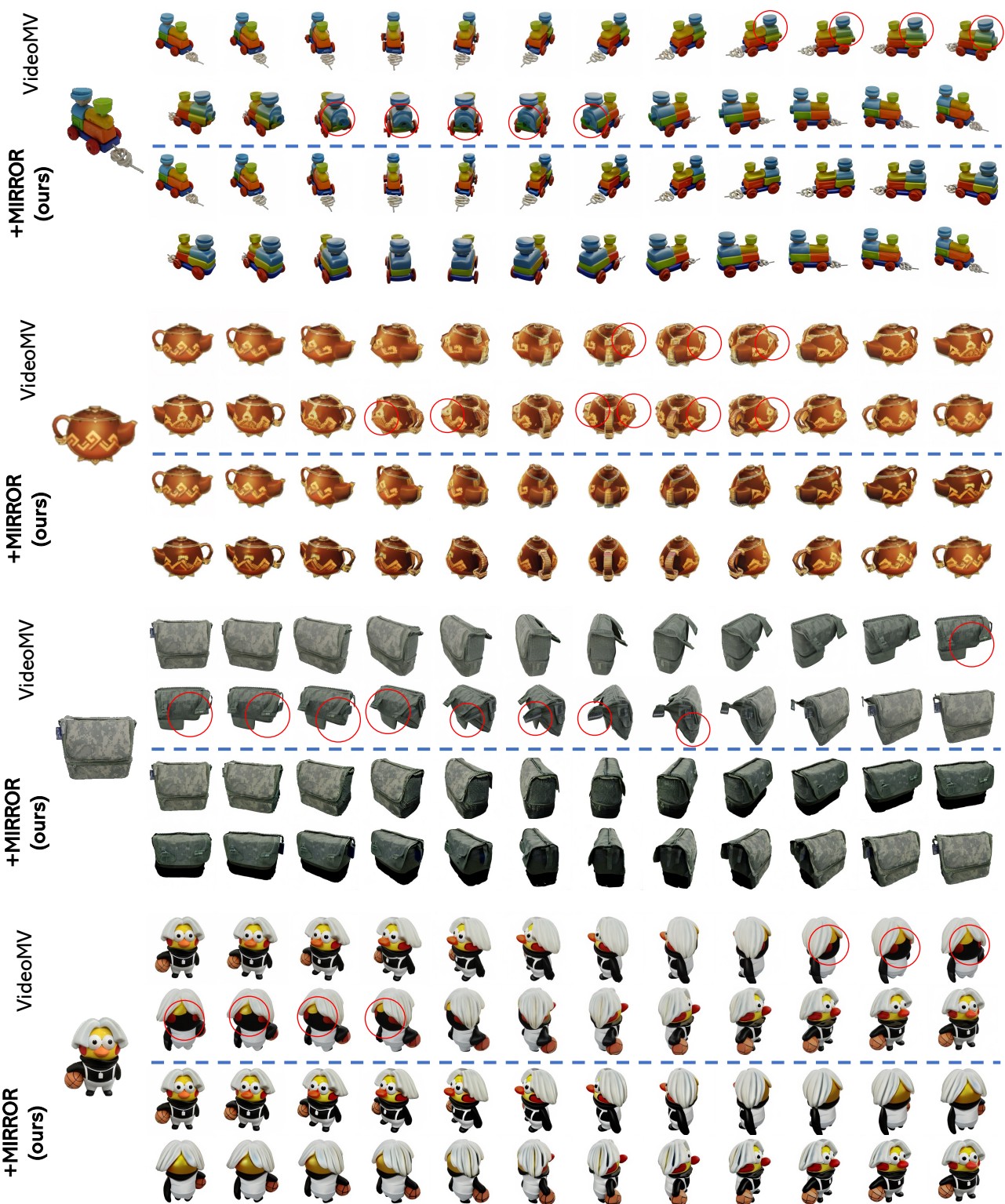

Figure 19: Generated multi-view images by applying MIRROR on VideoMV for image-based diffusion rectification. The areas with the Janus Problem or content drifting are marked with red circles.

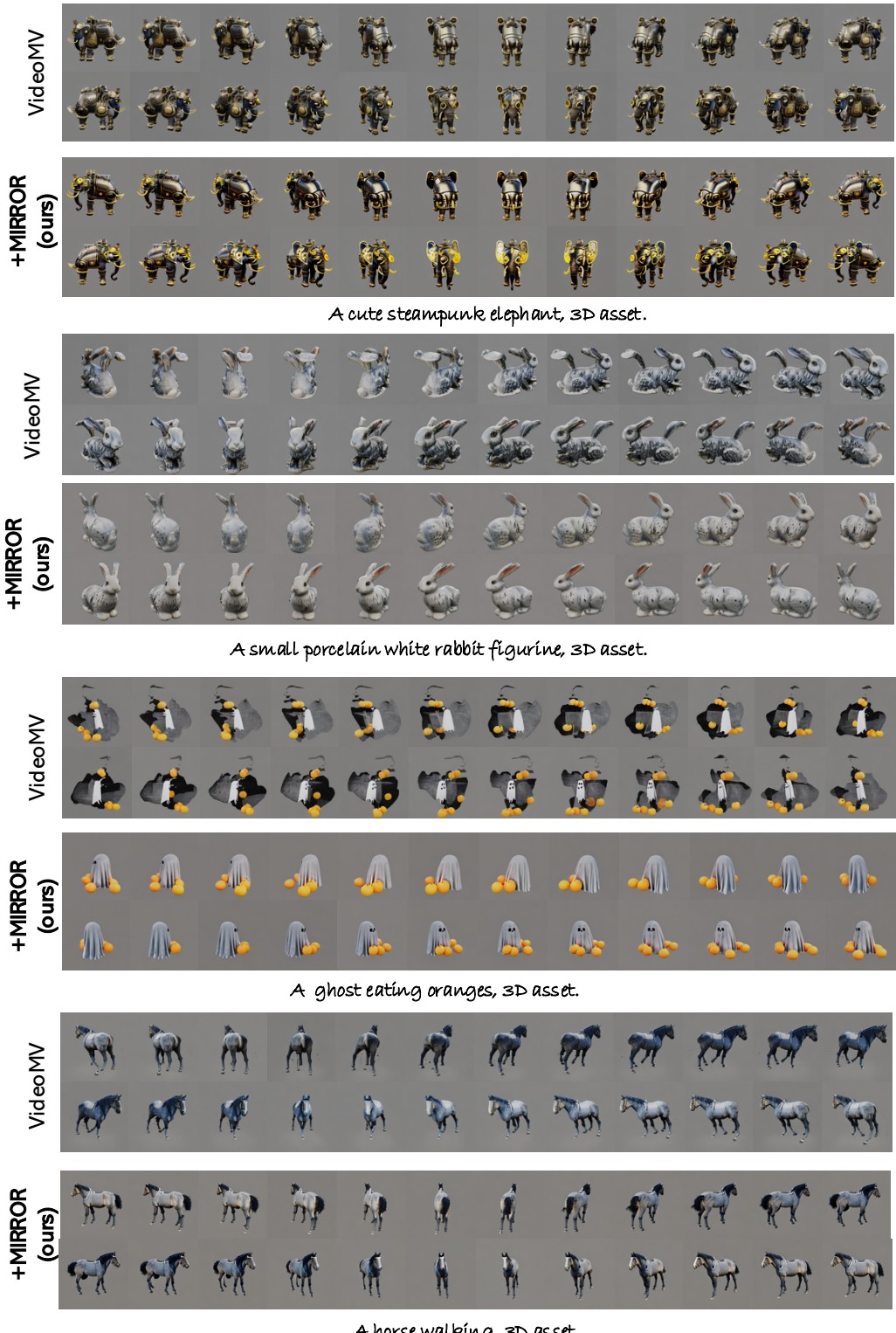

Figure 20: Generated multi-view images by applying MIRROR on VideoMV for text-based diffusion rectification.

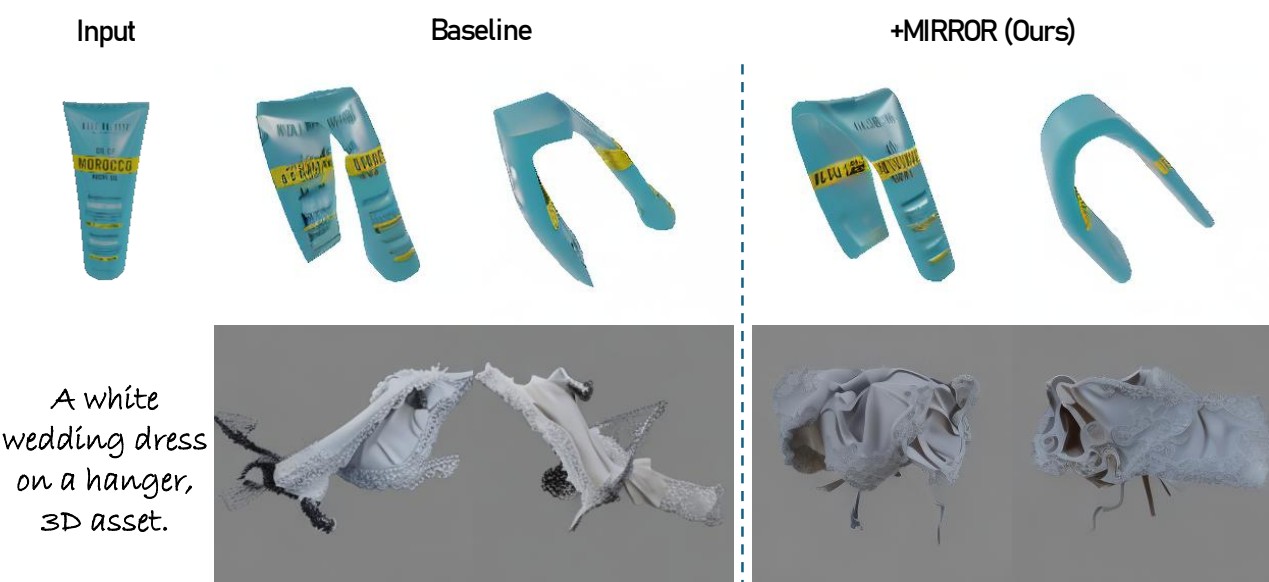

Figure 21: Failure Cases for both image-based (SyncDreamer) and text-based (VideoMV) models.

