# OpenReview forum: "MIRROR: Make Your Object-Level Multi-View Generation More Consistent with Training-Free Rectification"
_ICML.cc/2025/Conference — ICML 2025 poster_

### Official Review · Reviewer_ir3S · 2025-03-10

**Overall Recommendation:** 2

**Summary:**

This paper introduced  MIRROR, as a training-free rectification to improve the consistency of multi-view generation. The main contributions can be divided into (1) Trajectory Tracking Module (TTM) to pixel-wise trajectory tracking that labels identical points across views and (2) Feature Rectification Module (FRM) for explicit adjustment of each pixel embedding on noisy synthesized images by minimizing the feature distance. The overall idea is interesting, but the presentation of this paper is very unclear, while some details should be further clarified.

**Claims And Evidence:**

No. After reading the paper, I still could not understand why Trajectory Tracking Module (TTM) works with monocular depth estimation (depthanythingV2) without any metric alignment. The monocular depth is scale-invariable. Unlike the metric depth, monocular depth fails to be directly used as the condition of geometric warping.

**Essential References Not Discussed:**

No

**Experimental Designs Or Analyses:**

Yes. Most results are based on qualitative comparisons.

**Methods And Evaluation Criteria:**

Yes. This paper proposed both qualitative and quantitative results based on various base methods, showing the effectiveness.

**Other Comments Or Suggestions:**

N/A

**Other Strengths And Weaknesses:**

Except for the issue of TTM mentioned above. The presentation of this paper is also unclear, especially Sec4.3 is very hard to follow.
Many symbols are defined, but their usage is not clearly discussed. For example, Line231(right) defines the 3x3 block as $M(u)$, which has completely not been used and mentioned in the subsequent paragraphs at all. Moreover, what are the meanings of  $Z_{\alpha}$, $v\in B_u$. How to understand the feature distance of FRM (which symbol indicates block feature)?

**Questions For Authors:**

The authors should clarify about the usage of monocular depth in TTM, and clear and detailed presentation about Sec.4.3. It would be beneficial if the authors could present a clearer overview pipeline, while most symbols are labeled in this pipeline.

Besides, there is another question: why the invalid background information should be explicitly excluded during the feature information fusion, even if the TTM is correct? Is this problem caused by the incorrect depth warping, resulting in mistakenly relating to background regions? Would this limit the extension to scene-level multi-view generation?

**Relation To Broader Scientific Literature:**

The proposed method is a general approach to improve the consistency of multi-view diffusion models. But the discussion of this paper is limited to the object level with simple backgrounds, failing to be extended to the scene level.

**Theoretical Claims:**

This paper includes some theoretical claims. However, the claim of Eq.5 is questionable, i.e., $t\rightarrow 0$ leads to the convergence of the trajectory tracking operator, because the depth is not aligned as the metric depth. Even the depth is extracted from $x_0$, it still fails to be used as the warping condition.

---

> ### Author Rebuttal · Authors · 2025-04-01
>
> Thank you for your thorough analysis and constructive feedback on our paper. We will address the concerns you raised and hope our responses will clarify your doubts.
>
> ***Q1.  Scale of Depth***
>
> A1. Based on the camera parameters of the base model, we approximate relative-to-metric depth conversion and achieve depth alignment across multiple views.
>
> As the object-camera distance is fixed during training, the model learns a consistent depth scale under circular camera poses. This scale factor allows transforming relative depth into absolute depth for cross-view alignment. We further apply grid search over scale factors and use dual-anchor fusion of block features to reduce potential scale shifts during inference. The transformed depth enables more accurate trajectory tracking in TTM via geometric warping.
>
> Details on Eq. (5) can be found in **Reviewer y23b, A3**. And alignment details of depth estimation will be included in the appendix of the revised version.
>
> ***Q2.  More Quantitative Results***
>
> A2. Based on your suggestion, we have provided additional quantitative evaluation results. Please refer to the table in **Reviewer BJCB, A1**, for details.
>
> ***Q3.  Extension to Scene-level Tasks***
>
> A3. Importantly, the nature and manifestations of inconsistency differ between object-level and scene-level multi-view generation tasks. Object-level inconsistency mainly arises from the lack of 3D structural modeling and consistency supervision, often manifesting as the Janus problem and content drifting. In contrast, scene-level tasks typically suffer from layout disarray and semantic drift due to the absence of structural representations and layout supervision. Thus, the underlying challenges are fundamentally different.
>
> Our task track focuses on object-level multi-view generation, a key branch in 3D generation, with the baseline models representing mainstream, state-of-the-art methods. Our motivation is to correct the inconsistencies in object-level base models via explicit consistency supervision in a lightweight, plug-and-play, training-free manner, thereby improving 3D reconstruction quality.
>
> While scene-level generation is another important branch with fundamentally different inconsistency issues, we believe our method can offer insights for advancing this direction. To adapt MIRROR to this task, future work could incorporate layout-aware priors or scene maps to handle the broader spatial context. We consider MIRROR a core foundational step toward such extensions, and potential directions for scene-level adaptation will be discussed in the appendix.
>
> ***Q4.  Symbols in FRM***
>
> A4. We corrected the typo by redefining the 3×3 block as $B_{u _ \alpha}$ instead of $M(u _ \alpha)$ to align with the notation used in subsequent sections. Following your suggestion, we clarify key symbols in FRM, along with the updated pipeline  (see Fig. 1 in https://anonymous.4open.science/r/mirror-A9B9/figs.pdf).
>
> $u, u_\alpha$ denote the coordinates of a point in the current view and its corresponding tracked point in the neighboring view, respectively. And $Z(u), Z_\alpha(u_\alpha)$ represent the feature values indexed by point $u$ and $u_\alpha$. By traversing all features ${Z_\alpha(v), v \in B_{u_\alpha}}$, within block $B_{u_\alpha}$, and applying the dual-anchor weights $W$, we obtain the aggregated block feature $\mathcal{M}(Z_\alpha(u_\alpha))$. The L2 feature distance between $Z(u)$ and $\mathcal{M}(Z_\alpha(u_\alpha))$ is then computed by Eq. (10) to form the consistency correction loss.
>
> ***Q5. Background Exclusion***
>
> A5. As mentioned in our response A1, irrelevant background information would not caused by depth warping. The design of negative anchors aims to exclude both the additional information from depth map downsampling into the latent space and the redundant or irrelevant signals arising from expanding point features into block-level form to preserve spatial continuity.
>
> This design does not hinder extension to scene-level tasks. The dual-anchor feature fusion mechanism effectively suppresses irrelevant information while enhancing the contribution of relevant features. Moreover, dual anchors and their weights can be adapted to different application scenarios for better generalization.
>
> We will incorporate the revisions into the paper and hope our responses could address your concerns.
>
> We sincerely believe that our work is deserving of acceptance, and we would be grateful if you could recognize the contributions we’ve made. We kindly hope that, you might consider raising the score accordingly. Thank you again for your thoughtful feedback and consideration!

---

> > ### Comment · Reviewer_ir3S · 2025-04-05
> >
> > Thank you for the rebuttal. I appreciate your response addressing my concerns regarding the presentation, background exclusion, and the extension to scene-level tasks. Given the authors' assertion that object-level and scene-level multi-view generation tasks are fundamentally different, I recommend that this distinction be clearly articulated in the title and abstract of the paper. This clarity will help convey that the focus of this work is solely on object-level multi-view generation, as the current phrasing may lead to misunderstandings.
> >
> > Additionally, the authors noted that the monocular depth is aligned using camera poses. Could you please provide more details on how this process is implemented? I noticed that this critical aspect was not discussed in the main paper. Specifically, is it achieved through a grid search? If so, this approach could come across as overly idealized, as it would imply strong prior knowledge, i.e., all camera distances are the same and share the same metric scale.

---

> > > ### Author Response · Authors · 2025-04-07
> > >
> > > Thank you again for your kind response and insightful suggestions. Following your advice, we will clarify the term “object-level” in both the title and abstract, and explicitly emphasize in the introduction that our task targets object-level multi-view generation to avoid potential misunderstanding.
> > >
> > > Additionally, we would like to further clarify the depth alignment process. The first step involves estimating the depth for each view using a depth estimator. Second, the estimated depth is normalized to the range [0,1] as relative depth. Third, the relative depth is then multiplied by the scale factor, defined as the ratio of the baseline model’s camera distance to the average relative depth, to obtain the absolute depth for the generated images. Moreover, it is important to note that the prior knowledge we rely on is entirely derived from the baseline models. Since different baseline models provide different camera distances, the resulting depth scales also vary. Accordingly, the grid search is conducted independently for each baseline rather than using a shared scale. Consequently, our method is not restricted to a unified depth scale.
> > >
> > > Thank you again for taking the time to read and comment on our work! We hope that this explanation helps to further address your concerns.

---

### Official Review · Reviewer_y23b · 2025-03-13

**Overall Recommendation:** 3

**Summary:**

The paper introduces MIRROR, a training-free, plug-and-play method that improves consistency in multi-view image generation using diffusion models. At its core, MIRROR uses two novel modules: the Trajectory Tracking Module (TTM), which pinpoints corresponding 3D points across views using depth maps, and the Feature Rectification Module (FRM), which aligns features during sampling to fix inconsistencies. Unlike methods that require fine-tuning, MIRROR works directly during inference, making it compatible with popular pre-trained models like SyncDreamer and VideoMV. Experiments show it effectively tackles the Janus problem and content drift while preserving photorealism, offering a lightweight solution for high-quality 3D generation.

**Claims And Evidence:**

- MIRROR improves multi-view consistency in diffusion-generated images.

Qualitative: Visual comparisons (Fig. 1, 5) show MIRROR resolves artifacts like the Janus problem (e.g., multiple faces) and content drift (e.g., misaligned geometry) in baselines (SyncDreamer, VideoMV).

Quantitative: Metrics (Table 1) confirm gains in PSNR (up to +3.15), SSIM (up to +0.084), and LPIPS (up to -0.091), indicating improved alignment and reduced perceptual inconsistency.

- MIRROR is a training-free, plug-and-play solution compatible with existing models.

Uses DDIM inversion and rectification during inference (Fig. 3), requiring no fine-tuning or architectural changes to baselines.

Applied successfully to diverse models (SyncDreamer, MVD-Fusion, VideoMV) for both image- and text-based tasks (Table 1, Fig. 5).

- Depth-guided trajectory tracking enables precise geometric alignment.

Removing depth guidance (Fig. 7) leads to erroneous correspondences (e.g., mismatched limbs on animals, distorted shapes).

Proposition 4.2 and Appendix D show tracking errors diminish as denoising progresses, ensuring stable rectification.

- Feature rectification achieves efficiency without sacrificing quality.

Omitting UNet Jacobian terms (Theorem 4.4) reduces inference time by ~50% (Table 2) with negligible performance loss.

Dual-anchor fusion (Fig. 8) filters background noise while retaining critical features, validated by improved SSIM/LPIPS (Table 5).

**Essential References Not Discussed:**

**Limitations**

The paper does not discuss concurrent training-free rectification techniques and inverse diffusion papers, such as ”Denoising Diffusion Restoration Models“，“SOLVING VIDEO INVERSE PROBLEMS USING IMAGE DIFFUSION MODELS”

The description of DreamFusion is not entirely accurate. While DreamFusion does require optimizing the 3D representation (e.g., NeRF or Gaussian Splatting), it does not require backpropagation through the diffusion network. This should be clarified.

Recent works like Dust3R (2024) or Mast3R (2024) could enhance correspondence tracking and depth estimation but are not discussed.

Methods like EscherNet (2024) or CAT3D (2024), which handle arbitrary 6DoF poses and multiview conditioning, are omitted.

**Experimental Designs Or Analyses:**

The comparison with baselines is thorough, covering multiple models and metrics.

The ablation studies (Fig. 6, 7, 8) effectively isolate contributions from TTM, FRM, and dual-anchor fusion.

The timing analysis (Table 2) provides strong evidence that MIRROR is computationally efficient.

**Concerns**

The proposed method only works with camera poses at the same elevation angle (Eq. 4, Section 4.2), which is quite limited. How about using more generic point tracking methods, such as optical flow or dedicated point trackers?

Will the current setting work with models that support arbitrary 6DoF poses and flexible numbers of views, such as EscherNet and CAT3D?

Depth estimation is done using a monocular method. Could recent multi-view depth estimation approaches (e.g., Dust3R, Mast3R) improve scale and 3D consistency?

**Methods And Evaluation Criteria:**

- Methods

The proposed method employs a two-stage rectification pipeline:

1. Utilizes off-the-shelf models (e.g., SyncDreamer, VideoMV) to synthesize initial multi-view images.

2. Rectification via TTM and FRM:

    Trajectory Tracking Module (TTM): Uses monocular depth estimation (Depth-Anything-V2) to establish 3D correspondences across views.

    Feature Rectification Module (FRM): Aligns pixel embeddings via dual-anchor fusion and gradient guidance, applied during DDIM sampling.

**Limitations:**

TTM assumes fixed elevation angles (Eq. 4), limiting applicability to rigid objects and predefined camera paths (e.g., azimuth-only rotations).

Evaluated only on object-centric models (image/text-to-multi-view). Applicability to multi-view conditioned (e.g., EscherNet, CAT3D) or scene-level methods (e.g., CameraCtrl, MotionCtrl) remains unverified.


- Evaluation

Datasets: GSO (image-based), T3Bench (text-based).

Metrics: PSNR, SSIM, LPIPS (multi-view consistency), CLIP score (text alignment).

Quantitative and qualitative results clearly demonstrate the improvement over initial baseline models.

**Other Comments Or Suggestions:**

How does MIRROR perform on real-world multi-view data? The evaluated data is either synthetic or under perfect lighting/imaging conditions.

How sensitive is MIRROR to depth estimation errors, and could improving depth estimation further enhance performance?

Why is the trajectory tracking method restricted to the same elevation? Could more flexible tracking methods be incorporated?

How does MIRROR handle dynamic scenes or deformable objects where depth varies non-rigidly across views, say there is pretrained model can do 4D NVS.

**Other Strengths And Weaknesses:**

**Concerns**

Reference image selection in the text-based method for tracking loss is unclear. In Fig. 5, applying MIRROR causes significant changes in text-based outputs compared to single-image-based results. Why?

Table 1 shows that text-based methods gain the most improvement. Why is this the case?

**Questions For Authors:**

Please see my questions above and all the **limitations** and **concerns** in each part.

**Relation To Broader Scientific Literature:**

MIRROR builds on multi-view diffusion models (e.g., SyncDreamer, VideoMV) and depth-guided correspondence tracking. It differentiates from epipolar geometry-based methods (Ye et al., 2024; Zhou & Tulsiani, 2023). It also relates to inverse diffusion problems and test time optimization methods.

**Theoretical Claims:**

The paper presents several theoretical justifications, particularly in:

- Trajectory tracking convergence analysis (Proposition 4.2).

The proof tries to show that tracking error is bounded by diffusion model errors.

- Gradient-based rectification (Theorem 4.4).

The derivation in Appendix B.3 shows that neglecting the UNet Jacobian term introduces bounded errors, allowing efficient rectification by skipping the diffusion model backpropagation.

**Concerns**

While the theoretical framework is conceptually sound, gaps in notation, unverified assumptions, and incomplete derivations weaken rigor. Addressing these would strengthen the theoretical foundation.

Equation (5) uses, which is not clearly defined in the main paper.

Equation (15) in Appendix B.1 suggests that depth error is always smaller than pixel error, but this is not rigorously proven.

Equation (19) contains typos, missing the conditional term . Also, the transition from Eq. (17) and (18) to (19) is not obvious, particularly in obtaining the coefficients. A more detailed derivation would improve clarity.

---

> ### Author Rebuttal · Authors · 2025-04-01
>
> We greatly appreciate your thorough and detailed review, offering valuable insights on methodology, theory, experiments, and scalability, and thank you for recognizing our work!
>
> ***Q1. Limitation of TTM***
>
> A1. (1) TTM is designed to ensuring uniform geometry coverage and is theoretically extendable to any pose. In fact, sampling multiple views around a fixed elevation outperforms arbitrary 6DoF poses for reconstruction by reducing occlusions and preserving overall geometry. Specifying a set of elevations and sampling around each can further enhance results.
>
> (2) The number of views is determined by baselines, providing flexibility and decoupling from MIRROR.
>
> (3) Optical flow is unsuitable for multi-view generation due to high computational overhead, while dedicated point trackers suffer from drift during viewpoint rotation, causing geometric and texture distortions.
>
> In contrast, TTM enables fast tracking with block-level spatial fusion, reducing errors. 3D metrics in **Reviewer BJCB, A1**, also confirm our effectiveness.
>
> ***Q2. Broader Applicability***
>
> A2. Our task focuses on object-level multi-view generation and reconstruction, a key branch of 3D generation, with baselines being powerful mainstream methods. Generation of real-world or dynamic scenes and deformable objects are other important branches, and we believe MIRROR could inspire advancements in these areas. More details refer to **Reviewer ir3S, A3**.
>
> ***Q3. Definition of Eq.(5)***
>
> A3.  Eq.(5) defines the tracking error upper bound as base model's sampling error, with $x_0$ and $\hat{x}_0(x_t, t)$ representing the true image and predicted image at state t, respectively.
>
> ***Q4. Proof of Eq.(15)***
>
> A4. Using the first-order Taylor expansion, we obtain:
> $$
> H(x_0)=H(\hat{x}_0(x_t,t))+\nabla H(x_0) (x_0-\hat{x}_0(x_t,t))+o(||x_0-\hat{x}_0(x_t,t)||).
> $$
> Here, $H$ is a pretrained ViT network with a continuous, bounded gradient that outputs scale-consistent absolute depth, showing that the depth-level error is of the same order as the pixel-level error:
> $$
> ||H(\hat{x}_0(x_t,t))-H(x_0)||\approx||\nabla H(x_0)(x_0-\hat{x}_0(x_t, t))||\simeq O(||x_0-\hat{x}_0(x_t,t)||).
> $$
> ***Q5. Derivation of Eq.(19)***
>
> A5. There was a typo in Eq.(18), now corrected as:
> $$
> \nabla_{z_t}\log p_\theta(z_t)=-\frac{1}{\sqrt{1-\overline{\alpha}_t}}\varepsilon _ \theta(z_t,t).\tag{18}
> $$
> Using Eq.(17) and (18), Eq.(19) is easily derived:
>
> $$
> \varepsilon_\theta(z_t,t,c)=\varepsilon_\theta(z_t,t)-\sqrt{1-\overline{\alpha}_{t}}\nabla _ {z_t} \log p _ \theta(c|z_t).\tag{19}
> $$
> ***Q6. Depth Estimation***
>
> A6. Fig.12 and Tab.3 shows improving depth estimation accuracy benefits MIRROR, but this does not necessarily mean that multi-view depth estimation methods are superior.
>
> Our goal is a lightweight plugin to enhance consistency. While methods like Dust3R and Mast3R improve robustness, they require constructing multi-view cost volumes or 3D reconstruction, leading to high memory usage (2GB vs. 100MB for DA2) and slow inference, which compromises our advantages. Besides, in monocular tasks, DA2 significantly outperforms Dust3R, indicating that multi-view methods, despite improving depth consistency, may exacerbate Janus Problem due to the accumulation of estimation errors during denoising.
>
> For depth scale consistency, refer to **Reviewer ir3S, A1**.
>
> ***Q7.  Selection of s(t)***
>
> A7. s(t) aims to match the consistency gradient magnitude with $\varepsilon_\theta$. With negligible differences across models, we use a general parameter, though adjustments per model are feasible.
>
> ***Q8.  Failure Cases and Limitation of Baselines***
>
> A8. Failure cases (see Fig.2 in https://anonymous.4open.science/r/mirror-A9B9/figs.pdf) show that when the baseline produces unreasonable, severely flawed geometry (an inherent limitation), we struggle to correct these fundamental issues.
>
> ***Q9.  Essential References***
>
> A9. Without training, ConsiStory enhances subject consistency in text-to-image generation by modifying the network with Subject-Driven Shared Attention and Feature Injection. However, it lacks plug-and-play compatibility with other models, limiting generalizability.
>
> Multi-view diffusion models generate consistent images from noise, while inverse diffusion infers the initial state from known outcomes, with distinct objectives.
>
> ***Q10.  DreamFusion***
>
> A10. You're right. We'll clarify it.
>
> ***Q11.  Text-based Results***
>
> A11. $x_0$ is the theoretical true image in tracking loss (5). As the text-based method lacks a reference image, VideoMV uses half of the views for reconstruction and rendering as pseudo-ground truth. Moreover, text-based tasks are more challenging and diverse than image-based ones, so small corrections in the denoising process have a stronger impact, highlighting MIRROR's power through both qualitative and quantitative improvements.
>
> Hope our responses address your concerns. A detailed revision will be in the appendix. Thanks again for your comprehensive feedback.

---

> > ### Comment · Reviewer_y23b · 2025-04-05
> >
> > Thanks for the author's reply. There are still several concerns that remain.
> >
> > Q1.
> > (1) The author claims their method is "theoretically extendable to any pose". A theoretical possibility without experimental support is not rigorous.
> > Also, the author claimed, "In fact, sampling multiple views around a fixed elevation outperforms arbitrary 6DoF poses for reconstruction by reducing occlusions and preserving overall geometry. " Why? It is known that fixed elevation views cannot cover the complete views of objects, especially for complex objects that have self-occlusions.
> > (2) Although the method is designed to be flexible across baseline models, all baselines in the paper are constrained to fixed viewpoints and fixed view counts. Thus, it remains unclear whether TTM truly generalizes to multi-view settings with arbitrary or sparse camera poses.
> > (3) The argument dismissing point trackers and optical flow lacks ablation or comparative experiments. In practice, advanced point tracking and optical flow methods can offer accurate pixel-level correspondences. The claim that TTM outperforms these approaches needs stronger empirical backing.
> >
> > Q2.
> > The authors claim potential applicability to broader domains such as dynamic or scene-level generation. However, as discussed in Q1, there is no evidence showing MIRROR's ability to work beyond fixed-object scenarios. I echo Reviewer ir3S’s suggestion that the scope of the method should be explicitly limited to object-level multi-view generation with fixed viewpoints unless further validation is provided.
> >
> > Q3-Q5
> > Please clearly define all symbols and terms used in Eq. (5) in the main paper. Some notations are introduced without explanation, which affects readability and reproducibility.
> >
> > Q6.
> > While the authors highlight the lightweight design of their depth estimation module, the argument that multi-view methods like Dust3R "may exacerbate the Janus problem" is somewhat speculative. Modern multi-view depth estimators can be efficient and may help enforce scale-consistent depth across views, a desirable property for multi-view consistency. A clearer comparative analysis—especially in terms of trade-offs between accuracy and resource usage—would strengthen this point.
> >
> > Q9.
> > Several recent works on multi-view diffusion that directly address consistency—both spatial and temporal—are highly relevant and should be acknowledged. It would also be beneficial to demonstrate how MIRROR could complement such models. Evaluating MIRROR in conjunction with multi-view diffusion baselines would make a stronger case for its general applicability.

---

> > > ### Author Response · Authors · 2025-04-09
> > >
> > > We sincerely appreciate your feedback once again! Based on it, the experiments are in https://anonymous.4open.science/r/mirror-2C15/v2.pdf with figures and tables.
> > >
> > > ***Q1. TTM***
> > >
> > > (1) We define a general TTM using azimuth $\alpha$ and elevation $\phi$:
> > > $$
> > > x'=(x-\frac{W}{2})\cos{\alpha}+\frac{W}{2}-((y-\frac{H}{2})\sin{\phi}+\frac{H}{2}+z\cos{\phi})\sin{\alpha}, y'=(y-\frac{H}{2})\cos{\phi}+\frac{H}{2}-z \sin{\phi}.
> > > $$
> > > We found that large viewpoint gaps in arbitrary 6DoF views reduce cross-view correlation, leading to higher errors and limited improvement. In contrast, the fixed elevation ($\phi = 0$) provides stronger inter-view correspondence and better performance, so we adopt this setting. Circular camera poses provide stable, uniform coverage with stronger view correlations, enhaning 3D reconstruction. Sampling at multiple elevations further recovers occluded regions. More results for multi-object and complex examples are shown in Fig.3.
> > >
> > > (2) We clarify that the adopted baselines allow adjustment of both the number and configuration of static camera views. For instance, SyncDreamer and VideoMV generate up to 16 and 24 views, respectively, with elevation ranges selected from [−10°, 40°] and [5°, 30°].
> > >
> > > As noted in (1), although the baselines use static cameras, they are sufficiently effective for multi-view generation and reconstruction tasks. SOTA models like SyncDreamer, VideoMV, MVDiffusion++, and SV3D all use static cameras for dense multi-view generation. We will explicitly state in the paper that our task focuses on dense multi-view generation from static cameras.
> > >
> > > (3)  We replace TTM with the recent point tracking method CoTracker3 for comparison. As shown in Tab.1, CoTracker3 brings limited improvement over the baseline, while TTM achieves greater gains. Fig.1 further shows that CoTracker3 fails to resolve Janus Problem, which TTM effectively mitigates, demonstrating its superiority in handling multi-view inconsistency.
> > >
> > > ***Q2. Task Clarification***
> > >
> > > A. We will clarify "object-level" in the title and abstract and emphasize that our task focuses on object-level multi-view generation with fixed elevation. Additionally, we aim to extend our core training-free rectification pipeline to scene-level domains in future work.
> > >
> > > ***Q3-Q5. Eq.(5)***
> > >
> > > A. We provide a detailed definition of Eq.(5), where $\mathcal{T}_\alpha$ is the trajectory tracking operator (Definition 4.1), $x_t$ is the intermediate noisy state at time t, and $\hat{x}_0(x_t, t)$ is the predicted image at t:
> > >
> > > $$
> > > \hat{x}_0(x_t, t)= \frac{x_t-\sqrt{1-\overline{\alpha} _ {t}} \varepsilon _ \theta(x_t,t)}{\sqrt{\overline{\alpha}_t}},
> > > $$
> > >
> > > with $\varepsilon_\theta$ representing the noise prediction network, $x_0$ as the ground-truth image, and $O$ as an infinitesimal of the same order, $\Vert\cdot\Vert$ denotes the L2 norm. The clarified definition will be in the main paper.
> > >
> > > ***Q6. Comparison with DUSt3R***
> > >
> > > A. We replaced DA2 with the multi-view depth estimator DUSt3R. As shown in Tab.2, DUSt3R does not significantly increase inference time but requires more memory, whereas DA2 incurs minimal overhead. In low-memory environments (e.g., a single NVIDIA 3090 GPU), DUSt3R fails to run. While DUSt3R slightly improves PSNR and LPIPS, SSIM and CLIP Score remain comparable to DA2. Visual results in Fig.2 show marginal improvement, but in some cases, DUSt3R underperforms DA2.
> > >
> > > In summary, multi-view depth provides modest gains but with memory trade-offs. As depth estimation is independent of our core contribution, these results—along with those in Appendix E—show that MIRROR can benefit from ongoing advances in depth estimation.
> > >
> > > ***Q9. Discussion of Multi-view Models***
> > >
> > > A. Multi-view diffusion models, first introduced by MVDream, jointly train on 2D and 3D data for multi-view generation. To address inconsistency, several models employ strategies for both spatial and temporal alignment. SyncDreamer uses a 3D-aware attention mechanism for spatial consistency, MVD-Fusion employs noise-level depth estimations for reprojection, and VideoMV enhances spatiotemporal consistency with strong frame-to-frame coherence from video diffusion models. While these methods improve consistency, they lack explicit geometric constraints, entirely relying on learned networks. With limited 3D data, they are prone to local optima, and issues like Janus Problem and content drifting remain at inference.
> > >
> > > Methods like Consistent-1-to-3 and Era3D impose geometric constraints via epipolar geometry in multi-view attention, but Fig.4 in the main paper shows that epipolar correspondence leads to noisy supervision, causing over-smoothing and multi-face artifacts.
> > >
> > > Building on these efforts, MIRROR provides a training-free, efficient consistency enhancement, demonstrating significant effectiveness and generality across four SOTA diffusion models in the paper.
> > > As time constraints, we will explore additional models in future.
> > >
> > > Your comments are extremely helpful to us! Thank you.

---

### Official Review · Reviewer_BJCB · 2025-03-13

**Overall Recommendation:** 3

**Summary:**

The author present MIRROR, an efficient, training-free, plug-and-play method to enhance multi-view consistency in 3D asset generation. The proposed approach directly rectifies latent noisy images across views during the sampling process. To be specific, a Trajectory Tracking Module based on depth information is proposed to ascertain corresponding positions of points across distinct views. It use a Feature Rectification Module to eliminate ambiguity by enforcing consistency in the representation of the same physical point across different viewpoints. Qualitative and quantitative experiments demonstrate that MIRROR consistently enhances the performance of various generators.

**Claims And Evidence:**

The major claim of the proposed approach to be efficient, training-free, plug-and-play and enhance multi-view consistency is well supported by experimental results.

**Essential References Not Discussed:**

Essential references are well-discussed.

**Experimental Designs Or Analyses:**

1. The author use multiple current multi-view generation approaches as baselines and compare the generation results with and without enhancement using the proposed approach. The experiment results showing that incorporating MIRROR into these baselines resolves the prominent artifacts and multi-face issues encountered in baselines.

2. Comprehensive experimental analysis and ablation studies are give to demonstrate proposed approach's effectiveness and help understand how it works.

3.  The authors use PSNR, SSIM, LPIPS, and Clip Score as metrics for consistency. I have a concern on whether they can serve as effective metrics.

**Methods And Evaluation Criteria:**

Utilizing point-to-point corrections to ensure point features' consistency makes sense. Using the similarity computed between the feature of each pixel to provide a gradient map is an elegant way to guide the denoising procedure.

**Other Comments Or Suggestions:**

n/a

**Other Strengths And Weaknesses:**

The implementation and experiments details are well-described.

**Questions For Authors:**

1. What version of Depth-Anything-V2 is used? Does it output metric depth or relative depth? Any discussions on the depth scale consistency among multiple views?

**Relation To Broader Scientific Literature:**

This approach can serve as a plug-and-play module to the literature of multi-view generation, which may make impacts.

**Theoretical Claims:**

I check the proofs for the two major module and they look correct to me.

---

> ### Author Rebuttal · Authors · 2025-04-01
>
> We sincerely appreciate your valuable suggestions and the recognition of our work in methodology, theoretical proof, and experimental design. Below are our responses to your concerns, and we hope they help clarify any doubts you may have.
>
> ***Q1. More Metrics***
>
> A1. On the one hand, for fairness, we adopt the same quantitative metrics as baselines (including PSNR, SSIM, LPIPS, and CLIP Score), which are also widely used in current practice of multi-view generation. The quantitative results align well with the qualitative observations, further supporting the reliability of these metrics. In addition, **Reviewer y23b** acknowledges the quantitative metrics we used in the fourth line of their review in the **"Claims and Evidence"**.
>
> On the other hand, we additionally employ Chamfer Distance and Volume IoU to evaluate the 3D consistency of the reconstructed geometry from the generated multi-view images as follows:
>
>
> | Models | Chamfer Distance↓ | Volume IoU↑ |
> | -------------------- | ----------------- | ----------- |
> | SyncDreamer          | 0.0415            | 0.5137      |
> | **+MIRROR (ours)**   | **0.0387**        | **0.5296**  |
> | VideoMV (text-based) | 0.0459            | 0.5381      |
> | **+MIRROR (ours)**   | **0.0276**        | **0.6264**  |
>
> For text-based methods, following VideoMV, we sample half of the views at regular intervals to reconstruct a pseudo-ground truth mesh for evaluation. Other evaluation settings follow those in Appendix C.4. The 3D reconstruction metrics in the table further validate the effectiveness of our method in improving the quality and consistency of both multi-view images and 3D reconstruction.
>
> ***Q2. Depth Estimation***
>
> A2. All versions of Depth-Anything-V2 (DV2) achieve over 95% accuracy, with no significant performance differences observed in our task. Considering both accuracy and model size, we adopt DV2-Small as the depth estimation module.
>
> Moreover, based on the fixed camera distance used during training of each baseline, we convert the relative depth to metric depth using a consistent scale factor. Specifically, the baseline model, trained and inferred under circular camera poses, inherently provides a fixed depth scale, which ensures cross-view consistency. This allows the relative depth predicted by DV2 to be reliably transformed into absolute depth, ensuring consistent metric depth across multiple views.
>
> We hope our response could address your concerns. Based on your suggestions, we will include additional metrics in the appendix. Thank you once again for your recognition of our work and providing such insightful recommendations!

---

> > ### Comment · Reviewer_BJCB · 2025-04-08
> >
> > After reviewing the other reviewers' comments and the authors' rebuttal, I have decided to downgrade my rating to 'weak accept.' While most of my original concerns have been addressed in the authors' responses, I align with the valid concerns raised by other reviewers. Specifically:
> >
> > 1. While I believe focusing on object-level settings is reasonable, I concur with Reviewer ir3S’s suggestion that the scope of the method should be explicitly limited to object-level multi-view generation with fixed viewpoints.
> >
> > 2. I would like to see additional comparisons with methods based on multi-view depth estimation or point tracking.
> >
> > 3. I am interested in a discussion of recent works addressing the consistency of multi-view diffusion and would like to see comparisons between the proposed approach and these recent methods, or results from their conjunctions.

---

> > > ### Author Response · Authors · 2025-04-09
> > >
> > > Thank you for your attention to these aspects. We hope our responses would address your concerns.
> > >
> > > ***Q1. Task Clarification***
> > >
> > > A. We will clarify the term “object-level” in both the title and abstract of the paper, and explicitly emphasize in the introduction that our task focuses on object-level multi-view generation with fixed elevation to avoid potential misunderstanding.
> > >
> > > ***Q2. Comparisons with Point Tracking or Multi-view Depth Estimation***
> > >
> > > A. The experiment results are presented in https://anonymous.4open.science/r/mirror-2C15/v2.pdf, along with figures and tables.
> > >
> > > **(1) Comparisons with Point Tracking Method:**
> > >
> > > We incorporate the recent point tracking SOTA method, CoTracker3, as a substitute for TTM for comparison. Tab.1 shows that while CoTracker3 improves the baseline’s generation quality to some extent, the gains are less significant than those achieved with TTM. Moreover, Fig.1 shows CoTracker3 fails to resolve multi-face artifacts, exhibiting noticeable inconsistencies such as multiple legs and misaligned heads, which TTM effectively mitigates. Both quantitative and qualitative results demonstrate that TTM outperforms point tracking methods like CoTracker3 in addressing multi-view inconsistency.
> > >
> > > **(2) Comparisons with Multi-view Depth Estimation Method:**
> > >
> > > We replaced the original monocular depth model Depth-Anything V2 (DA2) with the multi-view model DUSt3R, which leverages neighboring views for reconstruction, thus providing multi-view depth.
> > >
> > > As shown in Tab.2, DUSt3R does not significantly increase inference time. But it requires substantially more memory, while DA2 incurs minimal overhead. In low-memory environments (e.g., a single NVIDIA 3090 GPU), DUSt3R fails to run. Besides, DUSt3R slightly improves PSNR and LPIPS, while SSIM and CLIP Score remain comparable to DA2. Visual results in Fig.2 show a slight improvement of DUSt3R, but in some cases, it underperforms DA2.
> > >
> > > In summary, multi-view depth can offer minor gains but comes with trade-offs in memory cost. Since depth estimation is a modular component independent of our core contribution in the pipeline, these results—along with those in Appendix E—demonstrate that MIRROR can continue to benefit from advances in depth estimation.
> > >
> > > ***Q3. Discussion with Consistent Multi-view Diffusion Models***
> > >
> > > A. Multi-view diffusion models were first introduced by MVDream, which jointly trains on 2D and 3D data for multi-view generation. To address inconsistency issues, several recent works incorporate strategies from both spatial and temporal perspectives. Specifically, SyncDreamer employs a 3D-aware attention mechanism to associate corresponding features across different viewpoints, enforcing spatial consistency. MVD-Fusion utilizes intermediate noise-level depth estimations for reprojection, also targeting spatial alignment for better 3D consistency. VideoMV further enhances spatiotemporal consistency by leveraging strong frame-to-frame coherence from video diffusion models. While these methods partially alleviate inconsistency, they lack explicit geometric constraints and rely entirely on learned networks. With limited 3D data, they are prone to local optima. As a result, issues such as Janus Problem and content drifting still persist at inference.
> > >
> > > Additionally, methods such as Consistent-1-to-3 and Era3D attempt to impose geometric constraints through epipolar geometry within multi-view attention to enhance consistency. However, we have demonstrated that epipolar correspondence provides noisy supervision, leading to over-smoothed results and multi-face artifacts, as shown in Fig. 4 of the main paper.
> > >
> > > Building on these efforts, MIRROR offers a training-free and efficient way to enhance generation consistency. Notably, we have demonstrated significant improvements in consistency across four SOTA diffusion models, validating its effectiveness and generality. Due to limited time, we will continue to explore additional multi-view diffusion models and include more experimental results in the appendix.
> > >
> > > We greatly appreciate your feedback, which has been extremely helpful in improving our work. We sincerely believe our work merits acceptance, and we kindly hope that, you might consider raising the score accordingly.

---

### Official Review · Reviewer_3noA · 2025-03-14

**Overall Recommendation:** 4

**Summary:**

This work introduces MIRROR, a training-free plug-and-play module designed to enhance the multi-view consistency of existing text-to-3D and image-to-3D diffusion models. In particular, MIRROR consists of two stages: the first stage leverages an off-the-shelf diffusion model to generate multi-view images, while the second stage obtains the corresponding noise via DDIM and applies feature rectification based on a trajectory tracking module to enhance multi-view consistency during the denoising process. The problem is formally defined and proved, and experiments conducted on several diffusion models verify the effectiveness of the proposed module.

**Claims And Evidence:**

Kindly refer to **Other Strengths And Weaknesses**

**Essential References Not Discussed:**

Kindly refer to **Other Strengths And Weaknesses**

**Experimental Designs Or Analyses:**

Kindly refer to **Other Strengths And Weaknesses**

**Methods And Evaluation Criteria:**

Kindly refer to **Other Strengths And Weaknesses**

**Other Comments Or Suggestions:**

N.A.

**Other Strengths And Weaknesses:**

## Strengths:

* The idea of enforcing consistency among adjacent frames to improve multi-view consistency is interesting and well-motivated.
* The problem is formally defined and well-proven.
* Experiments are conducted on several baseline models to demonstrate the versatility of the proposed module.

## Weaknesses:

* Unclear justification for multi-view diffusion limitations. It is unclear why multi-view diffusion alone cannot guarantee multi-view consistency, while the proposed regularization among adjacent frames can. Multi-view diffusion applies cross-view attention to enforce correspondence among generated views, which at a high level seems similar to the motivation behind the MIRROR module. A more detailed discussion and analysis of how FRM is more effective than cross-view attention in addressing multi-view consistency would be beneficial.

* Handling of non-Lambertian surfaces. Non-Lambertian (shiny) surfaces exhibit view-dependent effects, which contradict the underlying assumptions of MIRROR. It would be interesting to see how MIRROR will perform on assets with shiny surfaces.

* Impact of monocular depth inconsistency on TTM. Since the depth information is extracted from a monocular model, it cannot guarantee consistent scale and shift across different views. It would be interesting to know whether this issue affects overall performance. If so, would metric-depth or normalized scale-invariant depth be helpful in mitigating this problem?

**Questions For Authors:**

N.A.

**Relation To Broader Scientific Literature:**

Kindly refer to **Other Strengths And Weaknesses**

**Theoretical Claims:**

Kindly refer to **Other Strengths And Weaknesses**

---

> ### Author Rebuttal · Authors · 2025-04-01
>
> Thank you for your thoughtful feedback. We truly appreciate your acknowledgment of our motivation, methods, theoretical proof, and experimental results. Your encouraging comments are highly valued, and we are grateful for your insights!
>
> ***Q1. Comparison with Cross-view Attention***
>
> A1. Cross-view attention layers inherently struggle to enforce 3D consistency, as they lack explicit geometric constraints and rely solely on implicit learning through network weights. Given the limited availability of 3D data, such approaches are prone to local optima and often fail to generalize. Existing methods (e.g., MVDream, Consistent-1-to-3, Era3D) still suffer from issues like multi-face artifacts and content drifting at inference stage. Moreover, integrating our method into cross-view attention is impractical, as depth supervision is difficult to extract during training, and the attention layers demand substantial training costs in both computation and time.
>
> In contrast, FRM offers a more direct and effective solution. Leveraging the progressive denoising process of diffusion models, we introduce explicit multi-view consistency constraints by injecting expert priors from baseline models. It operates without requiring additional 3D supervision and is fully decoupled from the diffusion framework, making it lightweight, plug-and-play, and easily adaptable, which is a more novel approach at a high level. This explicit, geometry-aware mechanism enables FRM to correct inconsistencies more efficiently and reliably than cross-view attention, leading to more stable and coherent multi-view generation results.
>
> ***Q2. Handling of Non-Lambertian Surfaces***
>
> A2. This is a promising direction, and it could be addressed by decomposing the generation of non-Lambertian surfaces into two tasks.
>
> First, our pipeline enhances multi-view consistency to reconstruct high-quality 3D models, improving the geometric details of objects, which forms a solid foundation for subsequent lighting and rendering.
>
> Building on this, lighting models and physics-based rendering techniques can be applied to illuminate and render the 3D model, accurately capturing the reflective properties of shiny surfaces from various viewpoints.
>
> In other words, while MIRROR enhances the performance of the first task, it also supports the rendering of shiny surfaces. Together, the two processes improve the generation of non-Lambertian surfaces.
>
> ***Q3. Scale Consistency of Depth***
>
> A3. Although the depth information comes from the monocular estimation model, the depth ratio across different views remains consistent because the baseline model fixes the camera distance during training. Thus, the monocular model’s estimation capability is sufficient. Given that the depth scale perceived by the baseline model is fixed, we can use it as a scale factor to convert normalized relative depth into absolute depth, ensuring depth alignment across views.
>
> Furthermore, we perform a grid search for scale factors across multiple viewpoints and apply dual-anchor-based fusion of block features in FRM to mitigate potential scale shifts in TTM. We believe this approach is in line with your suggestion, and we will add these discussions in the appendix for further clarification.
>
> We hope our response has effectively addressed your concerns. Your suggestions have offered valuable insights that will greatly inform our future work, and we deeply appreciate them. Thank you once again for your constructive comments.

---

### Decision · Program_Chairs · 2025-05-01

**Decision:**

Accept (poster)

**Comment:**

The submission received 1 accept, 2 weak accepts and 1 weak reject. The reviewers generally found the proposed methodology to be well-motivated, interesting and designed elegantly. They also found the experiments to be comprehensive, with results validating the claims in the paper. One main concern was that the limitation to object-level multi-view generation was not adequately communicated, but in the rebuttal the authors have volunteered to define the narrower task more clearly in the title, abstract and introduction. There were other concerns about proofs of certain equations, comparison to other methods, and use of other metrics — these were effectively addressed by the authors in the rebuttal.

The AC concurs with the majority of the reviewers in recommending to accept this paper. The authors are requested to incorporate new and updated material from their rebuttals into the submission, as well as the agreed changes to more clearly convey the object-centric scope limitation of their work.